



# Advances in CALIPSO (IIR) cirrus cloud property retrievals – Part 1: Methods and testing

David L. Mitchell[1], Anne Garnier[2], and Sarah Woods[3]

[1] Desert Research Institute, Reno, NV 89512-1095, USA
[2][RSES]Analytical Mechanics Associates, Hampton, VA 23666, USA
[3] NSF NCAR, Boulder, CO 80301, USA

*Correspondence to*: David L. Mitchell (David.Mitchell@dri.edu)

**Abstract.** In this study, we describe an improved CALIPSO (Cloud-Aerosol Lidar and Infrared Pathfinder Satellite Observation) satellite retrieval which uses the Imaging Infrared Radiometer (IIR) and the CALIPSO lidar for retrievals of ice

particle number concentration $N_i$, effective diameter $D_e$, and ice water content IWC. By exploiting two IIR channels, this approach is fundamentally different than another satellite retrieval based on cloud radar and lidar that retrieves all three properties. A global retrieval scheme was developed using in situ observations from several field campaigns. The $N_i$ retrieval is formulated in terms of $N_i/A_{PSD}$ ratios, where $A_{PSD}$ is the directly measured area concentration of the ice particle size distribution (PSD), along with the absorption optical depth in two IIR channels and the equivalent cloud thickness seen by IIR.

It is sensitive to the shape of the PSD, which is accounted for, and uses a more accurate mass-dimension relationship relative to earlier work. The new retrieval is tested against corresponding cloud properties from the field campaigns used to develop this retrieval, as well as a recent cirrus cloud property climatology based on numerous field campaigns from around the world. In all cases, favourable agreement was found. This analysis indicated that $N_i$ varies as a function of cloud optical depth. By providing near closure to the ice PSD, the natural atmosphere may be used more like a laboratory for studying key processes

responsible for the evolution and life cycle of cirrus clouds and their impact on climate.

## 1    Introduction

Cirrus clouds contain only ice particles (i.e., no liquid cloud droplets), a condition guaranteed when cloud temperatures (T) are less than ~ -38° C (Koop et al., 2000). The microphysical and radiative properties of cirrus clouds are subject to very different ice nucleation pathways as well as whether the cirrus clouds are of liquid origin or not (e.g., Krämer et al., 2016), and

they also depend on aerosol particles of different sizes in complex ways (Ngo et al., 2024). With the ice particle size distribution (PSD) of cirrus clouds subject to so many factors, factors that may vary with latitude, season, and surface type (e.g., land vs. ocean), there is a need to observe cirrus cloud PSDs from space if cirrus clouds are to be represented accurately in climate models. If PSDs in cirrus clouds are approximated as exponential, they can be characterized through satellite retrievals of the PSD ice water content (IWC), effective diameter ($D_e$), and ice particle number concentration ($N_i$) as described





in Mitchell et al. (2020). Such satellite retrievals appear to be a necessary but not sufficient condition for understanding aerosol-cloud-climate interactions in cirrus clouds.

Ice crystals in cirrus clouds can form by either of two processes: homogeneous or heterogeneous ice nucleation (henceforth hom and het). The former requires no ice nucleating particles (INP) and can proceed through the freezing of haze and cloud solution droplets when $T \leq 235$ K (-38° C) and the relative humidity with respect to ice (RHi) exceeds some threshold where
RHi > ~ 145% (Koop et al., 2000). This results in generally higher concentrations of ice particles ($N_i$) relative to het (Barahona and Nenes, 2009; Jensen et al., 2012, 2013a,b; Cziczo et al., 2013). Under weak updraft conditions, $N_i$ resulting from hom may be similar to $N_i$ resulting from het (Krämer et al., 2016), and under atypical conditions (such as high concentrations of mineral dust), $N_i$ resulting from het can exceed 200 $L^{-1}$ which is characteristic of hom (Barahona and Nenes, 2009; Cziczo et al., 2013). In cirrus clouds, het may occur at any RHi > 100%, and in the context of a cloud parcel moving in an updraft, ice
is first produced through het, and subsequently through hom if the het-produced ice crystals do not prevent the RHi from reaching the threshold RHi needed for hom to occur (e.g., Haag et al., 2003). Overall, cirrus clouds formed primarily through hom will likely have substantially higher $N_i$ and IWC (due to the higher RHi of ice formation) relative to cirrus formed primarily through het (Krämer et al., 2016). Since the cirrus cloud extinction coefficient for sunlight is proportional to IWC/$D_e$, these two types of cirrus clouds (i.e., hom and het dominated cirrus) may therefore display considerably different radiative
properties.

In addition to extinction effects, relatively high $N_i$ produced through hom can result in smaller ice crystals that fall slower relative to het-formed ice crystals (Krämer et al., 2016). These lower ice fall speeds contribute to higher IWCs and longer cloud lifetimes, and thus greater cloud coverage (Mitchell et al., 2008). In this way hom alters cloud radiative properties through changes in $D_e$ and IWC (that affect cloud extinction and visible optical depth $\tau$) and also cloud coverage. Many
modelling studies have demonstrated the important impact that changes in ice fall speed have on climate (e.g., Sanderson et al., 2008; Mitchell et al., 2008; Eidhammer et al., 2017).

To date, there are two methods for retrieving all three cirrus cloud properties ($N_i$, $D_e$, IWC) from space: (1) the DARDAR approach based on the CloudSat Cloud Profiling Radar (CPR) and the CALIPSO (Cloud-Aerosol Lidar and Infrared Pathfinder Satellite Observation) satellite lidar CALIOP (Cloud and Aerosol Lidar with Orthogonal Polarization) as described in
Sourdeval et al. (2018) and Delanoë and Hogan (2010), and (2) a CALIPSO approach combining the Infrared Imaging Radiometer (IIR) with the CALIOP lidar as described in Mitchell et al. (2018; henceforth M2018). These two approaches differ in many respects, with the DARDAR approach sensing optically thicker clouds due to the CPR (i.e., the CALIPSO approach is limited to visible cloud optical depths $\tau < \sim 3$). But 79% of all ice clouds (of which cirrus clouds are a subcategory) have a $\tau < 3$ (Hong and Liu, 2015). Moreover, the DARDAR $N_i$ approach presumes a fixed PSD shape based on Delanoë et
al. (2014) whereas the CALIPSO approach does not assume a PSD shape, but rather is based on PSD properties obtained from aircraft measurement probes during cirrus cloud field campaigns. Both methods are sensitive to small ice crystals (that dominate $N_i$) due to the lidar regarding (1) and due to photon tunnelling (i.e., wave resonance) absorption regarding (2) which is most active when ice crystal lengths are comparable to the wavelength (~ 10 μm in this case) as described in M2018.





This study presents a new CALIPSO satellite retrieval that borrows some methodology from M2018 but also develops new methods that greatly increase the sampling range of cirrus clouds and increase the accuracy of the retrievals. It is similar to M2018 in that it retrieves $D_e$, $N_i$, and IWC by employing the effective absorption optical depth ratio, $\beta_{eff}$ [a standard, well characterized CALIPSO IIR retrieval using retrieved absorption optical depths at 12.05 µm and 10.6 µm in this case], but it differs in that new equations are used for calculating $N_i$, $D_e$ and IWC for greater accuracy and theoretical soundness as described in Sect. 2. As with M2018, empirical X - $\beta_{eff}$ relationships are developed from cirrus cloud field campaigns as described in Sect. 3, where X is a microphysical property such as $N_i/IWC$, but IWC is estimated more accurately and the retrieval is based on more field campaigns. Moreover, retrievals (and ice cloud radiative properties) at terrestrial wavelengths can be sensitive to the shape of the PSD as described in Mitchell (2002) and Mitchell et al. (2011). Such a sensitivity was found in the case of tropical tropopause layer (TTL) cirrus clouds, where their PSD shape differed from the anvil cirrus clouds sampled at higher temperatures. Due to this PSD shape difference, TTL and anvil cirrus having the same $\beta_{eff}$ can have different $D_e$, which was accounted for in this retrieval scheme. Finally, $\beta_{eff}$ was obtained with the most recent CALIPSO Version 4.51 Level 2 products. In Sect. 4, the retrievals are tested against corresponding cloud properties from the field campaigns used to develop this method, as well as the cirrus cloud property climatology of Krämer et al. (2020) based on numerous cirrus cloud field campaigns. Conclusions are given in Sect. 5. Scientific discoveries resulting from this CALIPSO retrieval are described in Part 2 of this study (Mitchell and Garnier, 2024).

## 2    Developing a new CALIPSO IIR retrieval for cirrus cloud properties

### 2.1 Analytical formulation

The retrieval of M2018 is based on co-located observations from the IIR and the CALIOP lidar aboard the CALIPSO polar orbiting satellite. It retrieves $N_i$, $D_e$ and IWC as a function of the effective absorption optical depth ratio $\beta_{eff}$, where $\beta_{eff}$ = $\tau_{abs}(12.05\ \mu m)/\tau_{abs}(10.6\ \mu m)$ and $\tau_{abs}(12.05\ um)$ and $\tau_{abs}(10.6\ \mu m)$ are the effective absorption optical depths retrieved in these IIR channels. $\beta_{eff}$ is considered an effective ratio since the retrieval of $\beta_{eff}$ from the cloud emissivity at each wavelength includes the effects of scattering. The $N_i$ retrieval depends on three empirical $\beta_{eff}$ relationships with $D_e$, the $N_i/IWC$ ratio, and the PSD effective absorption efficiency at 12 µm $Q_{abs,eff}$ (12 µm). These three $\beta_{eff}$ relationships were derived from in situ measurements during cirrus cloud aircraft field campaigns. The latter is used to derive the visible layer extinction, $\alpha_{ext}$, from $\tau_{abs}(12.05\ \mu m)$ and the IIR equivalent cloud thickness $\Delta z_{eq}$. $N_i$ is derived from the $N_i/IWC$ ratio after retrieving IWC from $\alpha_{ext}$ and the empirical $D_e$ - $\beta_{eff}$ relationship. The uncertainty in $N_i$ can be reduced by eliminating its dependence on the empirical $D_e$ - $\beta_{eff}$ relationship and by replacing the $N_i/IWC$ ratio with the $N_i/A_{PSD}$ ratio, where $A_{PSD}$ is the PSD projected area directly measured by the 2D-S probe (Lawson et al, 2006; Lawson, 2011). That is, the IWCs used to formulate the retrievals in M2018 were calculated from the ice particle projected area ($A_p$; directly measured by the 2D-S probe) and the ice particle mass (m) – $A_p$ power law relationship of Baker and Lawson (2006), where considerable uncertainty in IWC enters through this m – $A_p$ power law. This uncertainty can be eliminated by using the relationship between $N_i/A_{PSD}$ and $\beta_{eff}$, which can be determined



from PSD in situ measurements during cirrus cloud field campaigns, analogous to the calculation of the $N_i$/IWC ratio as described in M2018.

To remove these uncertainties, the retrieval of M2018 was reformulated as follows. We begin by equating two different expressions for the effective absorption coefficient $\alpha_{abs}$ for a homogeneous single-layer cirrus cloud:


$$\frac{\tau_{abs}(\lambda)}{\Delta z_{eq}} = \left[Q_{abs,eff}(\lambda)\right]_{\beta_{eff}} \left[\frac{A_{PSD}}{N_i}\right]_{\beta_{eff}} N_i \tag{1}$$

where $\tau_{abs}(\lambda)$ is the effective absorption optical depth at a given wavelength ($\lambda$), $Q_{abs,eff}(\lambda)$ is the PSD effective absorption efficiency ($\alpha_{abs}/A_{PSD}$) at the given wavelength and both $Q_{abs,eff}$ and $A_{PSD}/N_i$ are empirical functions of $\beta_{eff}$ (and are denoted accordingly). Moreover, $A_{PSD}$ is an area concentration, having units of area per unit volume, while $N_i$ is number per unit volume. This gives $A_{PSD}/N_i$ units of area per ice particle (e.g., $cm^2$). Solving for $N_i$, and applying to IIR channel at $\lambda = 12.05$

µm,

$$N_i = \left[\frac{N_i}{A_{PSD}}\right]_{\beta_{eff}} \frac{\tau_{abs}(12.05\ \mu m)}{\left[Q_{abs,eff}(12\ \mu m)\right]_{\beta_{eff}} \Delta z_{eq}} \tag{2}$$

Evaluating units, the rhs has units of reciprocal volume. As in M2018, $\beta_{eff}$ and $Q_{abs,eff}$ (12 µm) are based on in situ PSD measurements and the modified anomalous diffraction approximation (MADA) (Mitchell, 2000, 2002; Mitchell et al., 2006). Equation (2) is sensitive to the smallest ice crystals (which contribute the most to $N_i$) due to its dependence on $\beta_{eff}$, where $\beta_{eff}$

is sensitive to photon tunneling (i.e., wave resonance) absorption, and this type of absorption is strongest when ice particle size is comparable to the absorbed wavelength (e.g., M2018). The quantity $\Delta z_{eq}$ is smaller than $\Delta z$, the cloud layer geometrical thickness measured by CALIOP, and accounts for the fact that the IIR instrument does not equally sense all levels of the cloud layer that contribute to thermal emission. This is accounted for through the IIR weighting profile as discussed in M2018 and Garnier et al. (2021) and detailed later in Sect. 2.2.5.

The concept and definition of effective diameter $D_e$ is given in Mitchell (2002) as:

$$D_e = \frac{3}{2} \frac{IWC}{\rho_i A_{PSD}} \tag{3}$$

where $\rho_i$ is the bulk density of ice. This definition can be expanded to incorporate the $\beta_{eff}$ relationships pertaining to $N_i/A_{PSD}$ and IWC/$N_i$ (so that the $N_i$ terms cancel):

$$D_e = \frac{3}{2 \rho_i} \left[\frac{N_i}{A_{PSD}}\right]_{\beta_{eff}} \left[\frac{IWC}{N_i}\right]_{\beta_{eff}} \tag{4}$$

where the subscript $\beta_{eff}$ indicates that these ratios are retrieved quantities related to $\beta_{eff}$. Since the $D_e - \beta_{eff}$ relationship in M2018 was not as "tight" or precise as the $N_i$/IWC $- \beta_{eff}$ relationship, and the $N_i/A_{PSD} - \beta_{eff}$ relationship has a similar shape as the $N_i$/IWC $- \beta_{eff}$ relationship, Eq. (4) is expected to reduce uncertainties in the retrieval of $D_e$.



Cirrus cloud climatologies such as reported by Krämer et al. (2020) provide the spherical volume radius, $R_v$, of the mean ice particle mass, $IWC/N_i$. Unlike $D_e$, $R_v$ depends only on $IWC/N_i$ as

$$R_v = \left(\frac{3}{4\,\pi\,\rho_i}\right)^{1/3} \left[\frac{IWC}{N_i}\right]^{1/3}_{\beta_{eff}}. \tag{5}$$

With unique retrieval equations for $N_i$ and $D_e$, IWC is determined as

$$IWC = \frac{\rho_i}{3}\,\alpha_{ext}\,D_e \tag{6}$$

where $\alpha_{ext}$ is the shortwave or visible extinction coefficient given as

$$\alpha_{ext} = 2\left[\frac{1}{Q_{abs,eff}(12\,\mu m)}\right]_{\beta_{eff}} \frac{\tau_{abs}(12.05\,\mu m)}{\Delta z_{eq}}. \tag{7}$$

Similarly, the cloud visible optical depth is given as

$$\tau = 2\left[\frac{1}{Q_{abs,eff}(12\,\mu m)}\right]_{\beta_{eff}} \tau_{abs}(12.05\,\mu m) \tag{8}$$

and the cloud ice water path, IWP, is given as

$$IWP = \frac{\rho_i}{3}\,\tau\,D_e. \tag{9}$$

As noted, the relationships for the quantities $N_i/A_{PSD}$, $N_i/IWC$, and $Q_{abs,eff}(\lambda)$ related to $\beta_{eff}$ were derived from PSD measurements from cirrus cloud field campaigns. The field campaigns used here and in M2018 are the SPARTICUS (Small Particles in Cirrus) and TC4 (Tropical Composition, Cloud and Climate Coupling) field campaigns; see M2018 for details concerning SPARTICUS and TC4. The current study also uses the PSD measurements from the ATTREX and POSIDON field campaigns conducted in the tropical western Pacific, which are addressed in Sect. 2.3

## 2.2 CALIPSO processing and sampling improvements

The formulations presented above are applied to co-located CALIOP and IIR observations which provide $\tau_{abs}(12.05\,\mu m)$, $\beta_{eff}$, and $\Delta z_{eq}$ of cirrus cloud layers for selected scenes.

### 2.2.1 CALIPSO IIR data

While M2018 was using the version 3 (V3) CALIPSO products, this study uses the most recent version 4.51 (V4.51) products (Vaughan et al., 2024). The IIR Level 2 V4.51 track product reports cloud effective emissivities $\varepsilon_{eff}(12.05\,\mu m)$ and $\varepsilon_{eff}(10.6\,\mu m)$ at 12.05 µm and 10.6 µm at 1-km resolution, from which the respective effective absorption optical depths $\tau_{abs}$ are derived as (M2018, Garnier et al., 2021a)

$$\tau_{abs} = -ln\left(1 - \varepsilon_{eff}\right). \tag{10}$$



When both $\varepsilon_{eff}$ (12.05 µm) and $\varepsilon_{eff}$ (10.6 µm) are strictly between 0 and 1, $\beta_{eff}$ can be retrieved as

$$\beta_{eff} = \frac{\tau_{abs}(12.05\ \mu m)}{\tau_{abs}(10.6\ \mu m)}\ . \tag{11}$$

Both in M2018 and in this study, the calibrated and geo-located radiances are from the IIR version 2 Level 1 products (Garnier et al., 2018). The IIR effective emissivity retrievals are informed by CALIOP cloud detection and characterization as reported in the CALIOP V4.51 5-km cloud and aerosols layer products. IIR effective emissivities are similar in this study and in M2018 which was based on improved IIR V3 Level 2 data.

The contribution from the surface that enters in the computation of the effective emissivities was improved in the suite of
version 4 products after the analysis of IIR data in clear sky conditions as determined by CALIOP, following the same rationale as described in M2018. Land and oceans are first identified using International Geosphere and Biosphere Program (IGBP) surface types reported in the CALIPSO products. The presence of snow or sea-ice, which was based solely on a snow/ice index in M2018, is refined in version 4 by using the co-located 532 nm surface depolarization ratio reported by CALIOP. Following Lu et al. (2017), surface depolarization ratios larger than 0.6 are indicative of snow or sea-ice. Water, sea-ice, and snow types
are assigned different sets of static surface emissivities. Over snow-free land, the surface emissivity at 12.05 µm is also static, and the initial surface temperature provided as an input to the algorithm is adjusted to obtain radiative closure in clear air conditions. Surface emissivity at 10.6 µm is from in-house monthly daytime and nighttime maps (resolution: latitude x longitude = 1° x 2°) derived by again reconciling simulations and clear air observations.

The determination of the cloud radiative temperature, $T_r$, for the computation of the blackbody cloud radiance was improved
in version 4 following the rationale described in M2018. It is determined from the temperature at the centroid altitude of the CALIOP 532 nm attenuated backscatter profile and is further corrected using parameterized functions of emissivity and cloud thermal thickness (Garnier et al., 2021a).

In M2018, the atmospheric profiles and surface temperature used for the CALIOP and IIR retrievals were from the Global Modelling and Assimilation Office (GMAO) Goddard Earth Observing System version 5 (GEOS-5) model. In version 4, these
retrievals use the GMAO Modern-Era Retrospective analysis for Research and Applications version 2 (MERRA-2) model (Gelaro et al., 2017).

### 2.2.2 Cirrus cloud sampling

Because IIR is a passive instrument, we require, as in M2018, the cirrus cloud of interest to be the only cloud layer detected by CALIOP in the atmospheric column seen by the IIR pixel. Only clouds detected with a 5-km and 20-km horizontal averaging
of the CALIOP signal are considered. Importantly, IIR pixels containing clouds detected at the finest single shot (333 m) horizontal resolution are discarded (Garnier et al., 2021a). In addition, atmospheric columns where absorbing dust was detected by CALIOP are discarded. For this study, the identification of cirrus clouds relies on the CALIOP ice/water phase assignment of cloud layers, which was improved in version 4 (Avery et al., 2020). We select those clouds composed of Randomly Oriented Ice with high confidence in the phase assignment. In version 4, CALIOP cloud-aerosol discrimination is performed at any





altitude, whereas it was limited to the troposphere in version 3. Because of uncertainties in the determination of the tropopause altitude, upper troposphere tropical cirrus clouds were missed in version 3 but are included in version 4 (Fig. 15 of Avery et al., 2020). In addition, polar stratospheric clouds classified as ice are now sampled.

    We further require that the cirrus clouds are fully sampled by CALIOP to ensure that their true base is detected. These semi-transparent clouds that do not fully attenuate the CALIOP signal have an IIR effective emissivity at 12.05 µm smaller than
about 0.8 or visible optical depth smaller than about 3 (Fig. 2 of Garnier et al., 2021b).

    The radiative temperature is deemed representative of the IIR layer retrievals and for this study, we require $T_r$ to be colder than 235 K. Unlike in M2018, cirrus clouds with base altitude warmer than 235 K are included because of their CALIOP classification as ice is with high confidence.

### 2.2.3 Absorption optical depth uncertainties

Uncertainties in $\varepsilon_{eff}$ (12.05 µm) and $\varepsilon_{eff}$ (10.6 µm) induce uncertainties in $\tau_{abs}$(12.05 µm) and $\tau_{abs}$(10.6 µm) and subsequently in $\beta_{eff}$. In semi-transparent clouds, the main sources of error are from the measured radiances and from the surface contribution estimates, and errors increase as effective emissivity and optical depth decrease (M2018, Garnier et al. (2021a)). Errors in the surface contribution estimates are larger over land than over oceans due to the larger variability of surface emissivity and surface temperature over land. To evaluate IIR $\tau_{abs}$(12.05 µm), we use CALIOP 532 nm layer integrated attenuated backscatter
(IAB), which is an independent and measured quantity related to visible optical depth. Even though the relationship between CALIOP IAB and IIR $\tau_{abs}$(12.05 µm) depends on $Q_{abs,eff}$(12 µm), the extinction-to-backscatter lidar ratio, and the contribution of multiple scattering to the lidar backscatter (Garnier et al., 2015), CALIOP IAB is a reliable reference to assess IIR retrieval errors as optical depth and IAB tend to 0. Furthermore, CALIOP IAB uncertainties are not sensitive to land-oceans differences. Figures 1 and 2 show median IIR $\tau_{abs}$(12.05 µm) and percentiles vs. CALIOP IAB in six latitude bands over land and oceans,
respectively, during December-January-February (DJF) and June-July-August (JJA) of 2008, 2010, 2012, and 2013. The statistics are built using IAB bins of 5 x 10$^{-4}$ sr$^{-1}$ up to IAB = 0.02 sr$^{-1}$ including non-physical $\tau_{abs}$(12.05 µm) negative values resulting from retrieval errors. Because most of the samples have IAB > 5 x 10$^{-4}$ sr$^{-1}$, the lowest bin is from 5 x 10$^{-4}$ sr$^{-1}$ to 10$^{-3}$ sr$^{-1}$ where median IAB is ~ 7.6 x 10$^{-4}$ sr$^{-1}$. As CALIOP IAB tends to 0, median $\tau_{abs}$(12.05 µm) tends to zero as expected, both over land and over oceans. To IAB = 7.6 x 10$^{-4}$ sr$^{-1}$ corresponds a visible optical depth ($\tau$) ~ 0.016-0.026 assuming an extinction-
to-backscatter lidar ratio between 21 and 35 sr$^{-1}$ (Young et al., 2018), that is median $\tau_{abs}$(12.05 µm) ~ 0.0058-0.013 assuming $Q_{abs,eff}$ between 0.72 and 0.96 (see Eq. (8) and Sect. 3). The median $\tau_{abs}$(12.05 µm) values in the lowest bin reported in Table 1 are ~ 0.0091 ± 0.0054 over land and ~ 0.0065 ± 0.0013 over oceans. This is consistent with expectations and therefore shows no evidence of bias in the retrievals. Over land, the largest discrepancy is at 82° S- 60° S in DJF where median $\tau_{abs}$(12.05 µm) might be too large by ~ 0.01. Otherwise, the discrepancies are smaller than ± 0.003. Over oceans, the largest discrepancy is at
60° N-82° N in DJF where median $\tau_{abs}$(12.05 µm) might be too small by ~ 0.002. The spread of $\tau_{abs}$(12.05 µm) values at a given IAB is clearly larger over land in Fig. 1 than over oceans in Fig. 2 in the smaller range of IABs and up to IAB = 0.02 sr$^{-1}$ in winter in the polar regions, which is due to the larger IIR uncertainties over land resulting from the variability of surface





conditions. The $\tau_{abs}$ dispersions over land at high latitudes are about twice during winter relative to summer, which might be related to larger uncertainties in surface and atmospheric parameters and smaller radiative contrast between the surface and

the cloud temperature.

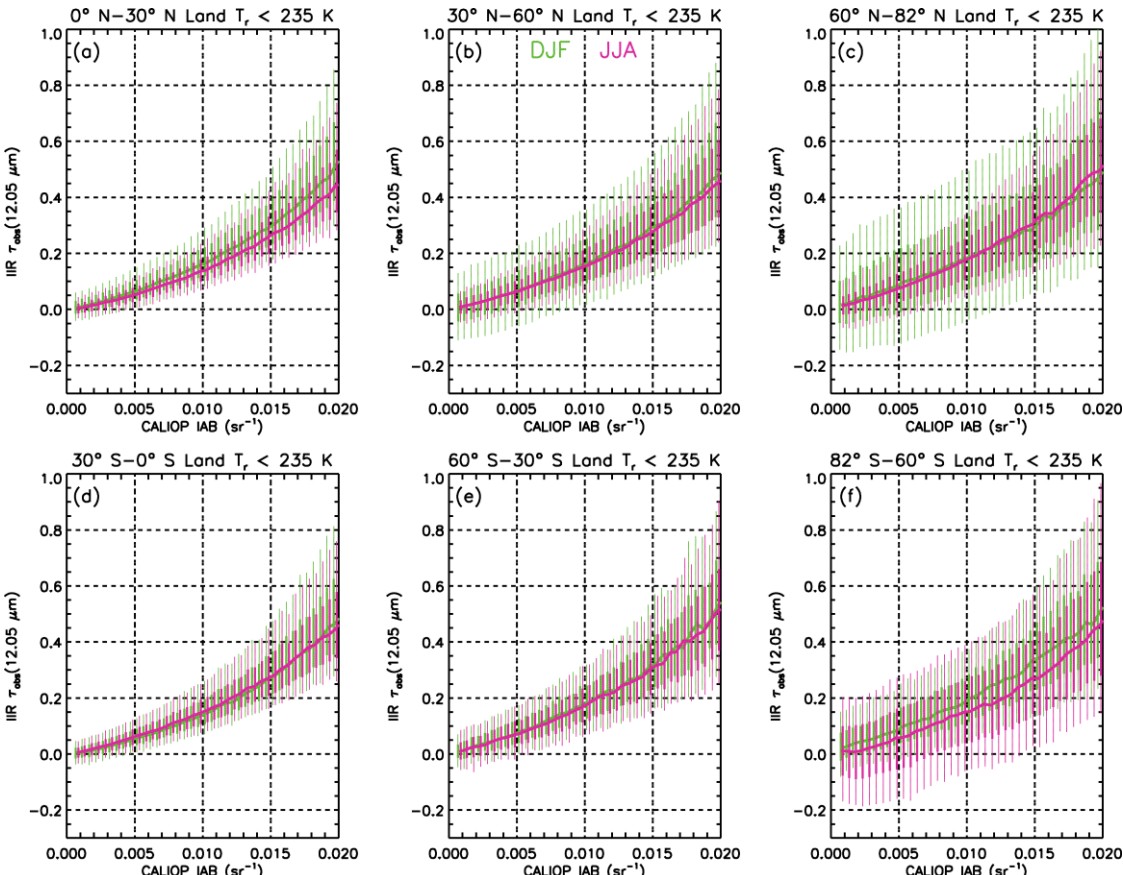

**Figure 1. IIR $\tau_{abs}$(12.05 μm) vs. CALIOP IAB over land in December-January-February (DJF, green) and in June-July-August (JJA, magenta). The solid curves show medians. The thin vertical lines are between the 10th and 90th percentiles and the**
**superimposed thick lines are between the 25th and 75th percentiles. Each row features the tropics (0-30°, panels a and d), midlatitudes (30-60°, panels b and e), and high latitudes (60-82°, panels c and f) in the northern (panels a-c) and in the southern (panels d-f) hemisphere during 2008, 2010, 2012 and 2013.**





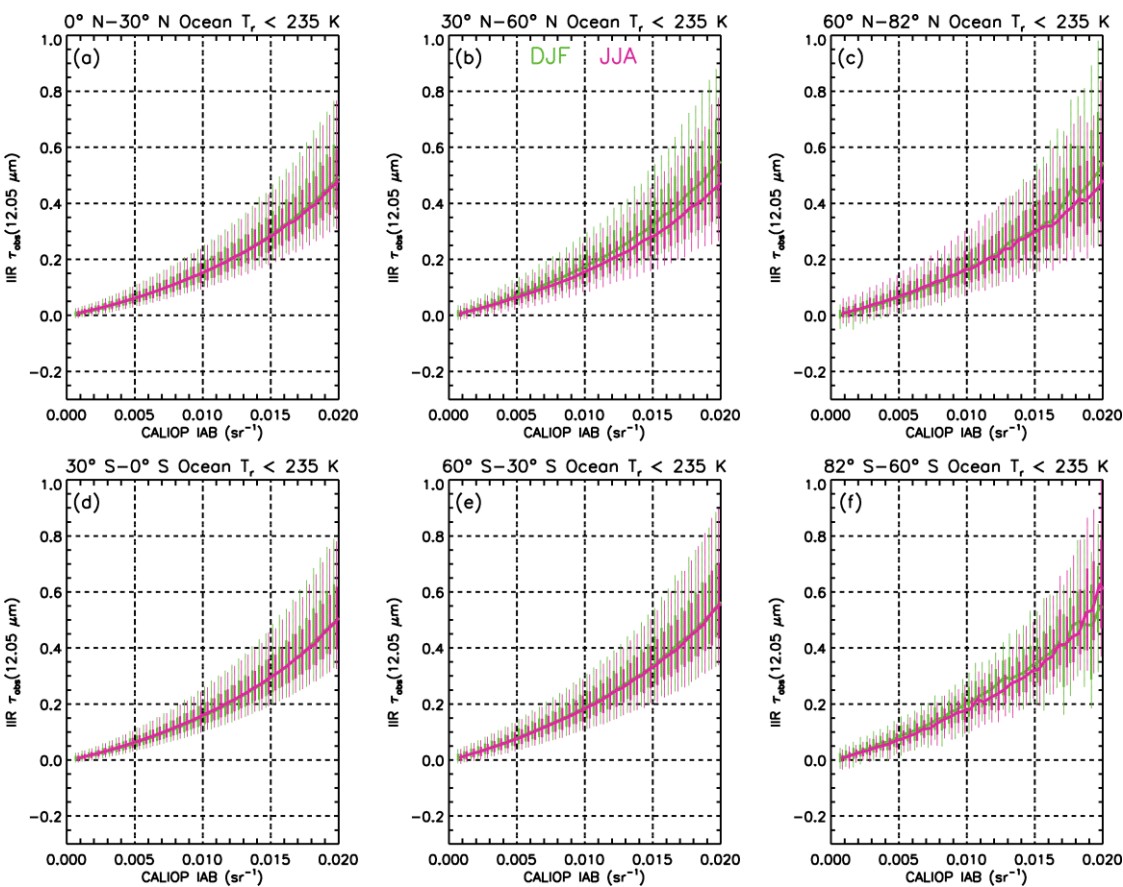

**Figure 2. Same as Fig. 1 but over oceans.**

Table 1: Median IIR $\tau_{abs}$(12.05 µm) in the lowest bin at CALIOP IAB ~ 7.6 x $10^{-4}$ sr[1] using all retrievals (cf Figs. 1 and 2).

| Latitude | Land | | Oceans | |
|---|---|---|---|---|
| | DJF | JJA | DJF | JJA |
| 60°-82° N | 0.0161 | 0.0116 | 0.0043 | 0.0088 |
| 30°-60° N | 0.0064 | 0.0076 | 0.0063 | 0.0062 |
| 0°-30° N | 0.0064 | 0.0032 | 0.0054 | 0.0071 |
| 30°-0° S | 0.0032 | 0.0047 | 0.0056 | 0.0059 |
| 60°-30° S | 0.0089 | 0.0081 | 0.0083 | 0.0081 |
| 82°-60° S | 0.0212 | 0.0119 | 0.0067 | 0.0056 |



### 2.2.4 Impact of optical depth uncertainties in $\beta_{eff}$ and measurement thresholds

Uncertainties in $\beta_{eff}$ are driven by optical depth uncertainties at 12.05 and 10.6 µm. In addition to the random noise, inter-channel biases of the retrievals could yield systematic biases in $\beta_{eff}$, which need to be assessed. A first approach is to evaluate the median $\tau_{abs}$(12.05 µm) - $\tau_{abs}$(10.6 µm) differences (hereafter $\tau_{abs}$12-10) when IAB tends to 0, i.e. in the lowest bin at CALIOP IAB ~ 7.6 x $10^{-4}$ $sr^{1}$ and using all retrievals. Since the imaginary index of refraction (a measure of absorption efficiency) at 10.6 µm is lower than at 12.05 µm, we expect positive differences. The results in Appendix A show that the

median differences are overall consistent with expectations, thereby showing no evidence of detectable biases.

However, $\beta_{eff}$ can be computed only when $\tau_{abs}$(12.05 µm) and $\tau_{abs}$(10.6 µm) have physical positive values. Therefore, because of retrieval random errors, especially over land, the $\tau_{abs}$(12.05 µm) and $\tau_{abs}$(10.6 µm) distributions are truncated when computing $\beta_{eff}$, because the scatter around median $\tau_{abs}$ at low IAB values will tend to be more negative at 10.6 µm relative to 12.05 µm. To illustrate the impact of these truncations, Fig. 3 shows the median $\tau_{abs}$(12.05 µm) - $\tau_{abs}$(10.6 µm) differences vs.

IAB over land for all retrievals (solid lines) and for samples having only positive values for both $\tau_{abs}$(12.05 µm) and $\tau_{abs}$(10.6 µm) (dashed line) for which $\beta_{eff}$ can be retrieved. These $\beta_{eff}$ values are also shown (diamonds) and are given by the right axis values. Figure 4 is the same as Fig. 3 but for retrievals over oceans.

The comparison of the solid and dashed lines in Fig. 3 over land shows that discarding non-physical absorption optical depths yields underestimated optical depth differences, and therefore underestimated $\beta_{eff}$ values. These systematic low biases

result largely from the greater percentage of negative values in the dispersion around $\tau_{abs}$(10.6 µm) at low IAB, so that using only the positive values yields a smaller (or negative) difference for $\tau_{abs}$(12.05 µm) minus $\tau_{abs}$(10.6 µm).

The greater separation between the solid and dashed curves in Fig. 3 during polar winter may relate to a lower contrast between the surface and cloud radiances and/or overall weaker radiances. But more generally, the divergence between these curves relates to the truncation bias noted above. Moreover, the decrease in $\beta_{eff}$ with decreasing IAB for IAB < 0.01 $sr^{-1}$ in

Fig. 3 in the Polar Regions, and IAB < 0.005 $sr^{-1}$ elsewhere, tends to roughly correspond with the divergence between the solid and dashed curves. These two trends are largely absent over oceans (cf Fig. 4) where surface emissivity and temperature are well characterized, thus greatly reducing the amount of scatter around the median values of $\tau_{abs}$(10.6 µm) and $\tau_{abs}$(12.05 µm). The only exception is at high latitude in the northern hemisphere (Fig. 4c) for both seasons where $\beta_{eff}$ at IAB smaller than 0.003 $sr^{-1}$ appears to be underestimated.

From this analysis, we chose a threshold IAB > 0.01 $sr^{-1}$ over land to ensure that the distributions are not or only slightly truncated. Nevertheless, for high latitudes in winter, median $\beta_{eff}$ is likely underestimated for IAB up to 0.02 $sr^{-1}$. The chosen IAB > 0.01 $sr^{-1}$ threshold corresponds to median $\tau_{abs}$(12.05 µm) ~ 0.15 (Figs. 1 and 2), that is $\tau$ > ~ 0.24-0.3 on average.

In M2018, an IAB threshold of 0.01 $sr^{-1}$ was applied to all retrievals, both over land and oceans. However, Fig. 4 shows that this condition can be relaxed over oceans. We refined the analysis over oceans by inspecting the fraction of negative $\tau_{abs}$(10.6

µm) values as $\tau_{abs}$(12.05 µm) increases from 0 with increments of 0.001. We estimate that the $\tau_{abs}$(10.6 µm) distribution is not significantly truncated when more than 90 % of the $\tau_{abs}$(10.6 µm) values are positive, yielding a lower threshold of 0.006 for



$\tau_{abs}$(12.05 µm) or $\tau > \sim 0.01$. The asterisks in Fig. 4 show that applying this threshold slightly increases median $\beta_{eff}$ at IAB $\leq$ 0.002 sr$^{-1}$, most notably in the tropics and at mid-latitude. Nevertheless, the width of the $\beta_{eff}$ distributions increases rapidly as IAB and optical depth approach zero, which is due in large part to increasing random uncertainties (Garnier et al., 2021b; M2018). This is illustrated in Table 2 for JJA at 0°-30°N, where the difference between the 75th and 25th $\beta_{eff}$ percentiles is ~ 0.49 for median $\tau_{abs}$(12.05 µm) = 0.02 but only ~ 0.07 for median $\tau_{abs}$(12.05 µm) = 0.49. The difference between the 90th and 10th percentiles is about twice these values.

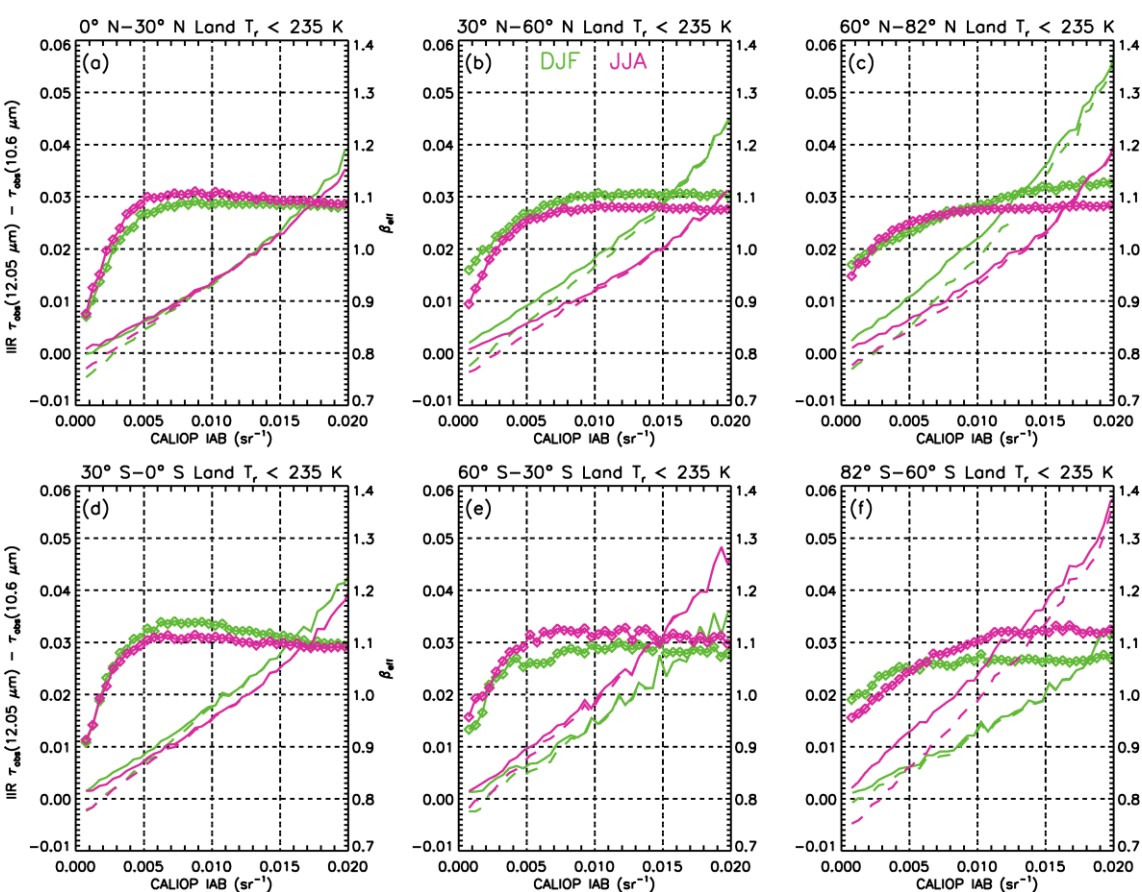

**Figure 3.** Median $\tau_{abs}$(12.05 µm) - $\tau_{abs}$(10.6 µm) difference vs. CALIOP IAB over land for all samples (solid), and for samples with positive absorption optical depths (dashed) for which $\beta_{eff}$ (diamonds, right vertical axis) can be retrieved. Each row features the tropics (0-30°, panels a and d), midlatitudes (30-60°, panels b and e), and high latitudes (60-82°, panels c and f) in the northern (panels a-c) and in the southern (panels d-f) hemisphere during December-January-February (DJF, green) and June-July-August (JJA, magenta) of 2008, 2010, 2012 and 2013.



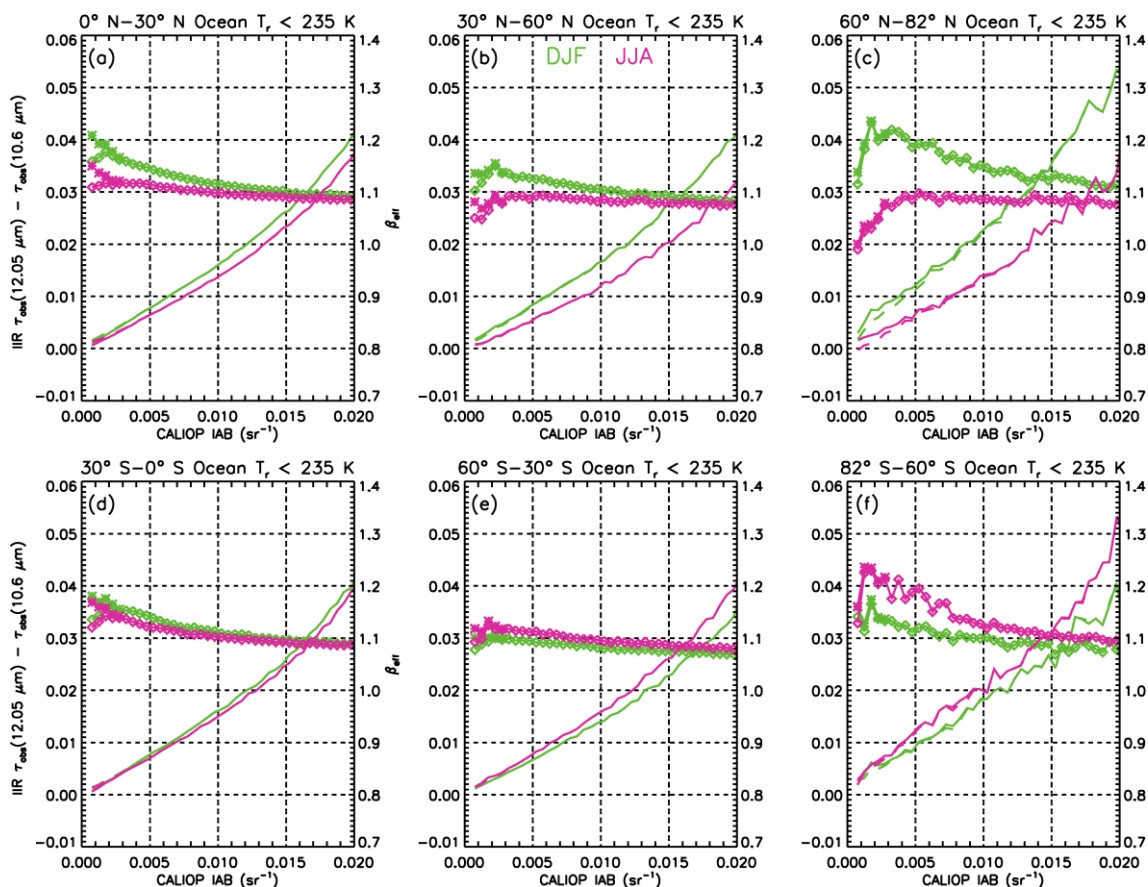

**Figure 4. Same as Fig. 3 but over oceans. In addition, the differences between the asterisks and the diamonds show that requiring $\tau_{abs}(12.05\ \mu m) \geq 0.006$ or visible optical depth $> \sim 0.01$ increases median $\beta_{eff}$ at IAB $\leq 0.002\ sr^{-1}$.**

Table 2: $\beta_{eff}$ percentiles, as well as 75th -25th and 90th-10th percentiles differences for three CALIOP IAB bins over oceans where $\tau_{abs}(12.05\ \mu m) > 0.006$ (or $\tau > \sim 0.01$).

| IAB | Median $\tau_{abs}$ | $\beta_{eff}$ | | | | | | |
|---|---|---|---|---|---|---|---|---|
| (sr⁻¹) | (12.05 µm) | 10th | 25th | median | 75th | 90th | 75th-25th | 90th-10th |
| 0.0008 | 0.02 | 0.765 | 0.943 | 1.150 | 1.431 | 1.946 | 0.488 | 1.181 |
| 0.0103 | 0.16 | 1.003 | 1.048 | 1.098 | 1.154 | 1.226 | 0.106 | 0.223 |
| 0.0202 | 0.49 | 1.025 | 1.054 | 1.086 | 1.122 | 1.164 | 0.068 | 0.139 |



### 2.2.5 IIR equivalent layer thickness, $\Delta z_{eq}$

As presented in Garnier et al. (2021a), the IIR equivalent layer thickness, $\Delta z_{eq}$, is estimated using the in-cloud 532 nm CALIOP
extinction profile of vertical resolution, $\delta z$, to build an IIR in-cloud weighting function and ultimately estimate the portion of
the cloud layer "seen" by IIR. For this analysis, we choose to use IIR channel centered at 12.05 µm.

The effective emissivity $\varepsilon_{eff}$ of a cloud composed of n vertical bins, i, from i = 1 at the cloud base to i = n at cloud top can
be seen as the vertically integrated in-cloud IIR effective emissivity attenuated profile $\varepsilon_{att}(i)$ written as:

$$\varepsilon_{att}(i) = \varepsilon(i) \prod_{j=i+1}^{j=n} \left(1 - \varepsilon(j)\right).$$
(12)

In Eq. (12), $\varepsilon(i)$ is the emissivity of bin i, and the second term represents the transmittance through the overlying cloudy bins.
The $\varepsilon(i)$ term is derived from the CALIOP cloud extinction coefficient of bin i, $\alpha_{part}(i)$, as

$$\varepsilon(i) = 1 - \exp\left(-\frac{\alpha_{part}(i)\,\delta z}{r}\right)$$
(13)

where r is the scaling ratio between the CALIOP layer optical depth, $\tau_{CAL}$, and the cloud effective absorption depth $\tau_{abs}$. The
IIR weighting function, $WF_{IIR}(i)$, is obtained from Eq. (12) after normalization by $\varepsilon_{eff}$ as

$$WF_{IIR}(i) = \frac{\varepsilon_{att}(i)}{\varepsilon_{eff}}$$
(14)

so that

$$\sum_i WF_{IIR}(i) = 1.$$
(15)

Then, we compute the IIR-weighted layer extinction coefficient, $\alpha_{CAL\text{-}IIR}$, as

$$\alpha_{CAL-IIR} = \sum_{i=1}^{i=n} \alpha_{part}(i)\, WF_{IIR}(i).$$
(16)

This IIR-weighted layer extinction coefficient is larger than or equal to the mean layer extinction, noted mean($\alpha_{CAL}$). Finally,
the IIR equivalent layer thickness, $\Delta z_{eq}$, is related to the geometric thickness $\Delta z$ as

$$\Delta z_{eq} = \Delta z \, \frac{mean(\alpha_{CAL})}{\alpha_{CAL-IIR}}.$$
(17)

The ratio r in Eq. (13) is estimated using CALIOP visible optical depth, and might thus differ from $2/Q_{abs,eff}$ (12 µm) used in
this study to derive visible optical depth, $\tau$ (Eq. (8)). Importantly, $\Delta z_{eq}$ does not depend on this ratio. Had we used $\tau$ instead of
$\tau_{CAL}$, both $\alpha_{part}$ and r in Eq. (13) would have been multiplied by $\tau/\tau_{CAL}$ and $\varepsilon(i)$ and the subsequent $WF_{IIR}(i)$ would have been
unchanged. Likewise, both mean($\alpha_{CAL}$) and $\alpha_{CAL\text{-}IIR}$ in Eq. (17) would have been multiplied by $\tau/\tau_{CAL}$, leaving $\Delta z_{eq}$ unchanged.
We note, however, that without vertically resolved information, $2/Q_{abs,eff}$ (12 µm) (or r) is supposed constant within the layer.

Two cirrus examples are shown in Fig. 5 where the CALIOP extinction coefficient ($\alpha_{part}$) profile is in black and the IIR
weighting function ($WF_{IIR}$) profile is in red. The vertical resolution is $\delta z = 0.06$ km. The first example in panel a is a TTL





cirrus between 15.13 and 16.5 km observed in June 2010. Retrieved $\varepsilon_{eff}$ is equal to 0.06 and the black and red curves have an almost identical shape. We find $\Delta z_{eq}$ = 0.52 km, which corresponds roughly to the main marked peak and to the secondary maximum. In panel b, the cirrus is between 6.74 and 10.74 km in the Southern Ocean in August 2008. Here, retrieved $\varepsilon_{eff}$ is equal to 0.44 and relative to the black curve, the lower part of the cloud contributes less to the cloud emissivity than the upper part. The equivalent thickness $\Delta z_{eq}$ = 2.98 km can be seen as the portion of the cloud above 7.8 km where $WF_{IIR}$ exceeds about

0.008. In these examples, $\Delta z_{eq}/\Delta z$ is equal to 0.38 (a) and 0.74 (b). Figure 6 shows $\Delta z_{eq}$ vs. $\Delta z$ for all the sampled cirrus over oceans, showing that $\Delta z_{eq}/\Delta z$ is globally mostly between 0.5 and 0.9.

The IIR weighting function as illustrated in Fig. 5 is also used to determine the cloud radiative temperature, $T_r$, (Garnier et al., 2021a), which is given in red in each panel. The temperatures in black are $T_{top}$ and $T_{base}$ at the layer top and base altitudes, respectively. In panel a where $WF_{IIR}$ exhibits one main peak, the altitude corresponding to $T_r$ is 15.9 km, near the $WF_{IIR}$

maximum. The $T_r$-$T_{top}$ difference is 45 % of the thermal thickness. In panel b, the altitude corresponding to $T_r$ is 9.1 km located between the two peaks of comparable amplitude, slightly closer to the upper one. $T_r$ is slightly closer to the top with a $T_r$-$T_{top}$ difference of 36 % of the thermal thickness. As discussed in M2018 and illustrated in Fig. 7 showing $T_r$-$T_{top}$ against $T_{base}$-$T_{top}$ for all the sampled cirrus over oceans, $T_r$-$T_{top}$ represents typically 30 to 70 % of $T_{base}$-$T_{top}$. Using temperature difference as a proxy for altitude difference, it appears that $T_r$ is on average at mid-cloud.


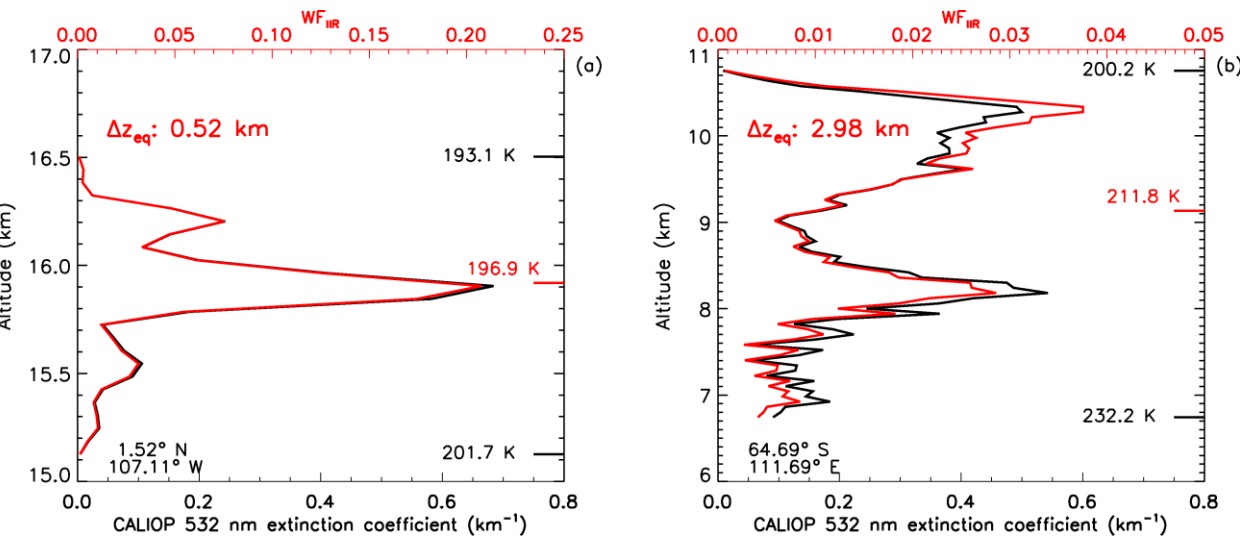

**Figure 5. Two cirrus cloud examples showing the CALIOP extinction coefficient ($\alpha_{part}$) profile in black and the IIR weighting function ($WF_{IIR}$) profile in red, with $\Delta z_{eq}$ indicated in red. The temperatures in black on the right-hand side of each panel are at cloud top and base, and in red is the radiative temperature ($T_r$) at the corresponding altitude. These examples are extracted from**
**CALIPSO granules (a) 2010-06-04T08-39-58ZN and (b) 2008-08-12T07-26-54ZD, with latitude and longitude given in the lower left corner of the respective panels.**



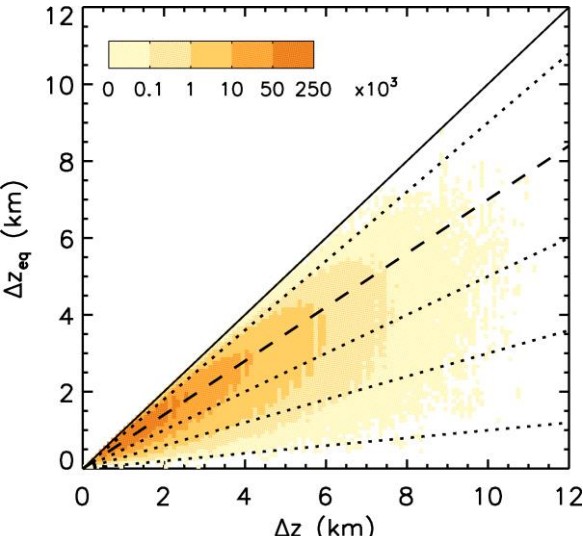

**Figure 6. $\Delta z_{eq}$ vs. $\Delta z$ for all sampled cirrus over oceans during 2008, 2010, 2012, and 2013. The colors represent the IIR pixels density. The dashed and dotted lines, from bottom to top, represent $\Delta z_{eq}/\Delta z$ of 0.1, 0.3, 0.5, 0.7, and 0.9.**

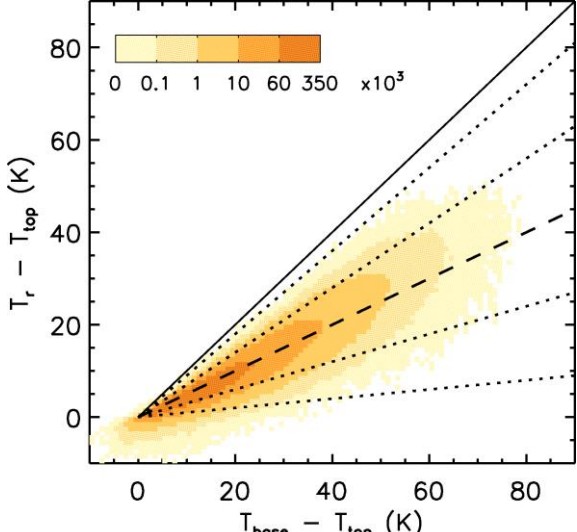

**Figure 7. $T_r$ - $T_{top}$ vs. $T_{base}$ - $T_{top}$ for all sampled cirrus over oceans during 2008, 2010, 2012, and 2013. The colors represent the IIR pixels density. The dashed and dotted lines, from bottom to top, represent $(T_r$ - $T_{top})/(T_{base}$ - $T_{top})$ of 0.1, 0.3, 0.5, 0.7, and 0.9. Using temperature difference as proxy for altitude difference, it appears that $T_r$ is on average at mid-cloud.**



## 2.3 Inclusion of additional tropical cirrus field campaigns

The M2018 CALIPSO retrieval was based on XX – $\beta_{eff}$ relationships (where XX refers to $N_i$/IWC, $D_e$, or $2/Q_{abs,eff}(12\mu m)$) developed from the SPARTICUS (Jensen et al., 2013a) and TC4 (Toon et al., 2010) cirrus cloud field campaigns. In this new
retrieval, the ATTREX and POSIDON cirrus cloud field campaigns (Jensen et al., 2017; Schoeberl et al., 2019) conducted in the tropical western Pacific were also used for this purpose, along with the SPARTICUS and TC4 field campaigns. Cirrus clouds were sampled in POSIDON by the NASA WB-57 aircraft, which flew the SPEC Inc. Fast Cloud Droplet Probe (FCDP; Glienke and Mei, 2020; Lawson et al., 2017), two-dimensional stereo (2D-S) probe (Lawson et al., 2006), and Cloud Particle Imager (CPI; Lawson et al., 2001). During ATTREX, they were sampled by the Global Hawk unmanned aircraft system, which
flew the SPEC Inc. Hawkeye instrument to measure ice PSDs between -50°C and -85°C but mostly in the tropical tropopause layer (TTL) between -65°C and -85°C (Woods et al., 2018). The Hawkeye houses three instruments (the FCDP, the 2D-S probe, and the CPI) that measure the complete cloud PSD and the corresponding size-resolved cloud particle shapes. The FCDP and 2D-S probe tips are designed to minimize ice particle shattering, and particle interarrival times are used to identify and remove clusters of particles resulting from shattering (Baker et al., 2009). The FCDP sampled particles between 1 and 50
μm while the 2D-S sampled ice particle maximum dimension D from 10 to 1280 μm (Woods et al., 2018), although D > 1280 μm can be estimated up to 4 mm (Jensen et al., 2017). However, the first (5-15 μm) size-bin of the 2D-S probe and the last size-bin (45-50 μm) of the FCDP were not used for producing composite mean PSDs. Figure 8 shows representative mean PSD examples from POSIDON (on left) and ATTREX (on right) along with information on corresponding effective radius ($R_{eff}$), $N_i$, and IWC. The agreement between the FCDP and 2DS probes where they overlap (from 15 to 45 μm, indicated by
the red and blue histograms) was good (as shown here) for most of the PSD measurements. The number of PSD during POSIDON having notably poorer agreement than those in Fig. 8 was 3 out of 66 PSDs in total, or 4.5%, with similar findings for ATTREX. Moreover, Jensen et al. (2013a) found good agreement for $N_i$ between the 2D-S and another $N_i$ probe (the Video Ice Particle Sampler or VIPS) when the first size-bin of the 2D-S probe was not considered.

For the SPARTICUS and TC4 campaigns, PSDs were measured only by the 2D-S probe. To determine whether ice particle
concentrations in the first size-bin (i.e., $N(D)_1$) of the 2D-S probe should be used for calculating the $N_i$/IWC – $\beta_{eff}$, $N_i$/$A_{PSD}$ – $\beta_{eff}$ and $1/Q_{abs,eff}(12\mu m)$ – $\beta_{eff}$ relationships from these campaigns (that were used in this retrieval as described in Sect. 3), PSDs from the POSIDON campaign were qualitatively evaluated from PSD plots provided by SPEC, Inc. The good agreement noted above between the FCDP and 2D-S probes from 15 μm to 45 μm suggests that the FCDP measurements from 1 to 15 μm may also be realistic. Jensen et al. (2013a) states that "In nearly all of the 2D-S size distributions, the concentration in the
first size bin (5-15 μm) is considerably larger than the concentrations in the next few larger bins, and the first bin often contributes significantly to the overall ice concentration." We found this to be true of the ATTREX-POSIDON 2D-S measurements as well. For the POSIDON PSDs, $N(D)_1$ of the 2D-S was within a factor of ~ 2.5 of the combined corresponding FCDP bins for 23% of the PSDs but exhibited much higher factors ranging from 3 to 32 for the other PSDs. On average, the 2D-S $N(D)_1$ was a factor of $10.4 \pm 8.1$ greater than the ice particle concentration in the corresponding FCDP bins. Therefore,





regarding the SPARTICUS and TC4 PSDs, we modified the measured PSDs by dividing the 2D-S $N(D)_1$ by 10.4 to approximately correct for this behaviour. While this correction would not always be valid for a single PSD measurement, it may be realistic for a large ensemble of PSDs. Relevant information for the SPARTICUS and TC4 field campaigns can be found in M2018. In M2018, different assumptions regarding $N(D)_1$ resulted in different retrieval formulations, but in the current approach, only one retrieval formulation is needed and presented.


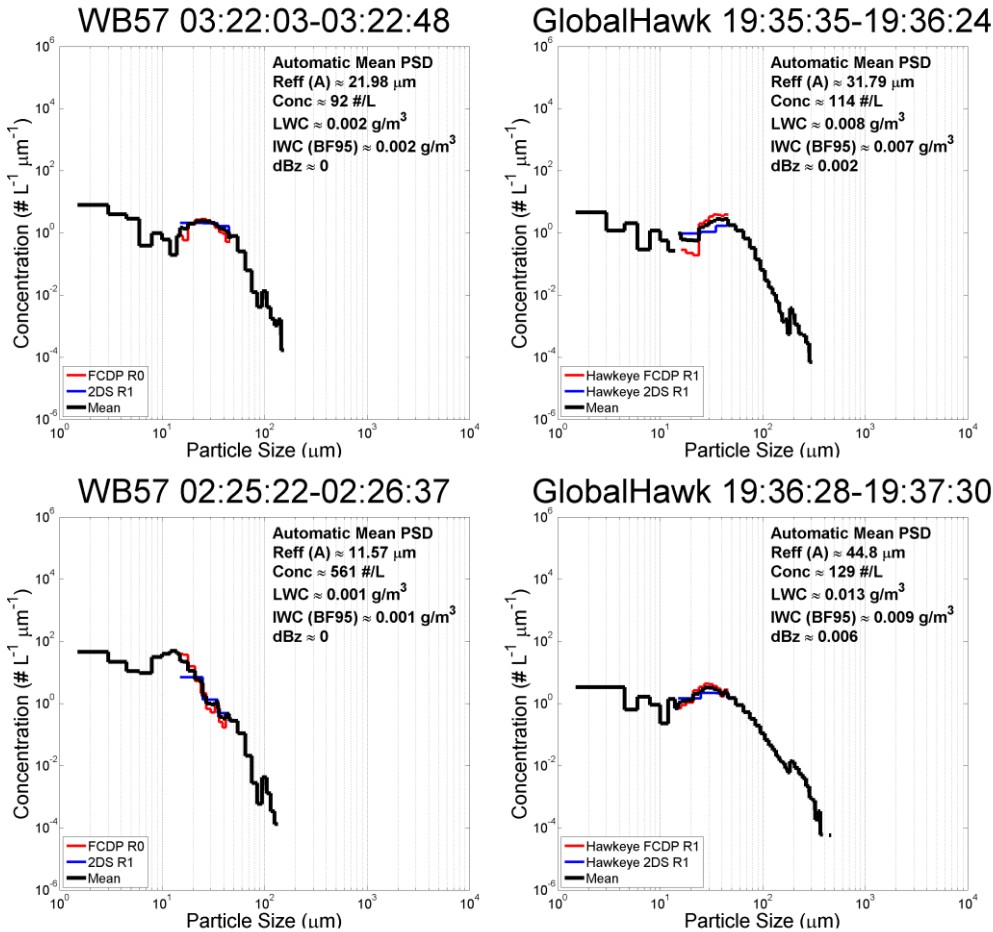

**Figure 8. Representative mean composite PSD from the ATTREX (right) and POSIDON (left) field campaigns, sampled by the FCDP (1.5 – 45 µm) and 2D-S (15 – 1280 µm) probes. The two probes in their overlap region (red and blue histograms) yield relatively consistent values, providing confidence in these measurements.**






## 2.4 Treatment of ice water content

As shown in Fig. 8, ATTREX and POSIDON PSDs were generally narrow, with maximum ice particle sizes generally less than 400 µm and often less than 150 µm. The Baker and Lawson (2006) ice particle area-mass expression that has normally been used to calculate the IWC for SPEC, Inc. PSD data predicts a spherical ice particle mass greater than predicted for spherical ice particles at bulk ice density (0.917 g cm⁻³) when ice particle maximum dimension D < 47 µm (which is non-physical). Since much of the PSD mass during the ATTREX and POSIDON campaigns is often associated with D < 47 µm, the ice particle mass-dimension expressions described in Erfani and Mitchell (2016; henceforth EM2016) were used for developing relationships in this retrieval scheme since the EM2016 mass-dimension relationships were designed to calculate the mass of small particle sizes < 100 µm more realistically. These EM2016 relationships are shown in Fig. 9, along with relationships from Lawson et al. (2019) for marine anvils cirrus, from Mitchell et al. (2010), and from Weitzel et al. (2020). It is seen that these relationships are relatively consistent, especially for D < 100 µm where uncertainties are greatest.

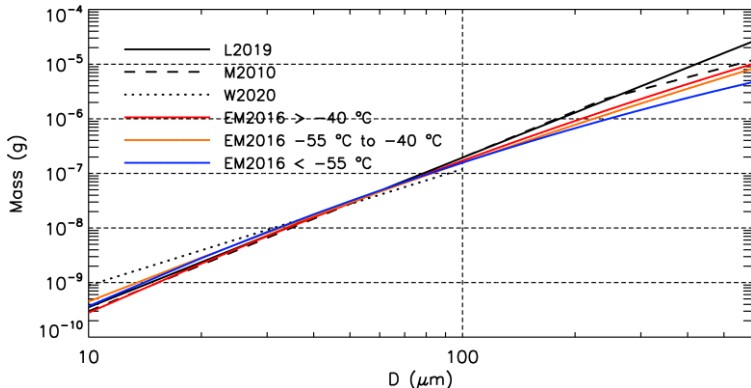

**Figure 9. Relationships between mass m (g) and particle dimension D (µm) from Lawson et al. (2019) (L2019, solid black line), Mitchell et al. (2010) (M2010, dashed black line), Weitzel et al. (W2020, dotted black line), and from EM2016 for temperatures colder than 55 °C (blue), between -55 °C and -40 °C (orange) and warmer than -40 °C (red).**

### 2.4.1 Mass-dimension relationship and $\beta_{eff}$

We recall that $\beta_{eff}$ of the particle size distribution is the ratio of effective absorption efficiencies at 12.05 and 10.6 µm, where 'effective' refers to the scattering contribution (see Eqs, 4 and 5 of M2018). For this discussion, we can assume that $\beta_{eff} \sim \beta$, i.e. the ratio of absorption efficiencies at 12.05 and 10.6 µm. As discussed in Mitchell et al. (2002), the relevant dimension to characterize the absorption efficiency of the single particle at a given wavelength, λ, is the effective distance $d_e$ defined as

$$d_e = \frac{m}{\rho_i \cdot A_p} \; . \tag{18}$$

Likewise, absorption efficiencies, $q_{abs}$ (λ), derived from MADA are uniquely related to $d_e$, as shown in Fig. 10a for the IIR channels using several ATTREX and POSIDON PSDs. The X-axis is (3/2) x $d_e$, noting that this quantity is the effective





diameter of the single particle. While $A_p$ is directly measured, m is derived from mass-dimension or mass-area relationships, so that $d_e$ depends on these relationships. The discontinuities in $q_{abs}(\lambda)$ result from changes in the MADA tunneling (i.e., resonance) efficiencies that depend on ice particle shape and size (M2018; Sect. 2.3). The PSD absorption efficiency $Q_{abs}(\lambda)$ is obtained after integration of $q_{abs}(\lambda)$ over the area distribution, $A(D)$, where D is particle dimension. Because $q_{abs}(\lambda)$ is

uniquely related to $d_e$, $Q_{abs}(\lambda)$ can be written

$$Q_{abs}(\lambda) = \int_{PSD} q_{abs}(\lambda, d_e(D)) \ A(D) \ dD \tag{19}$$

and

$$\beta = \frac{Q_{abs}(12 \ \mu m)}{Q_{abs}(10.6 \ \mu m)} . \tag{20}$$

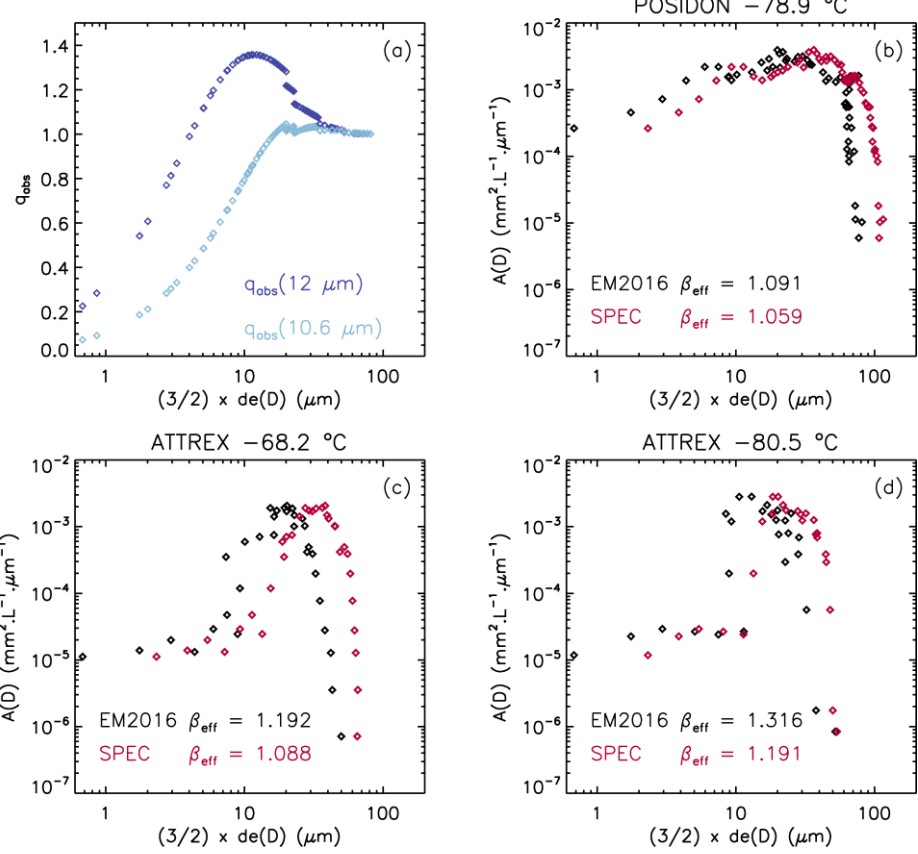


**Figure 10.** Panel (a) shows the unique relationship between absorption efficiency at 12 μm (dark blue) and 10.6 μm (light blue) and effective distance $d_e(D)$. Panels (b-d) show three examples of A(D) vs. (3/2) x $d_e(D)$ from the POSIDON (b) and the ATTREX (c-d) campaigns with $d_e(D)$ computed using particle mass from EM2016 (black) and SPEC (red). The smaller mass using EM2016 yields smaller $d_e(D)$ and larger $\beta_{eff}$ values.






It appears that $Q_{abs}(\lambda)$ and $\beta$ of a PSD depend on the variation of A(D) with $d_e$, which again depends on the estimated mass of the single particle. This is illustrated in Figs. 10b-d with three examples from the ATTREX and POSIDON campaigns, where the same A(D) is shown vs. (3/2) x $d_e$ using the mass from EM2016 in black and from SPEC in red. The black distributions are shifted towards smaller $d_e$ values compared to the red ones, yielding a larger $\beta$ (and ultimately $\beta_{eff}$) value using the EM2016 relationships. In these examples, the $\beta_{eff}$ values are increased from 1.059 to 1.091 in panel a, from 1.088 to 1.192 in panel b, and from 1.191 to 1.316 in panel c.

### 2.4.2 IIR $\beta_{eff}$ – temperature comparisons with SPEC and EM2016

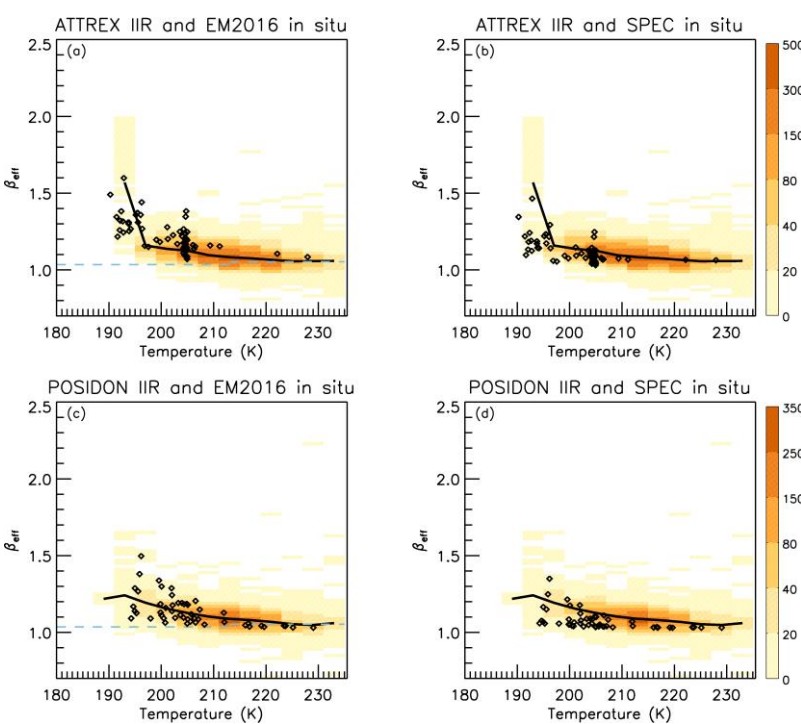

**Figure 11. IIR $\beta_{eff}$ vs. temperature in 0° N - 20° N and 130° E - 160° E during February and March 2014 (ATTREX, panels and b) and October 2016 (POSIDON, panels c and d) compared with $\beta_{eff}$ of PSDs (diamonds) measured during the denoted campaigns using the EM2016 (panels a and c) and the SPEC (panels b and d) mass-dimension relationships. The colors indicate IIR samples density and black curves represent median IIR $\beta_{eff}$. The horizontal dashed light blue lines in panels a and c indicate $\beta_{eff}$ at the sensitivity limit (see Sect. 3). IIR optical depth > ~ 0.3.**

To assess the impact of the m-D relationships on in situ $\beta_{eff}$, we compared $\beta_{eff}$ of PSDs measured during the ATTREX (2014) and POSIDON campaigns with independent IIR $\beta_{eff}$ retrievals (Fig. 11). Because one-to-one comparisons are not possible, we compared $\beta_{eff}$ vs. temperature, which is layer radiative temperature for IIR. To match the field campaigns, IIR samples are in 0° N - 20° N and 130° E - 160° E during February and March 2014 for ATTREX and October 2016 for POSIDON (Schoeberl et al., 2019). Comparisons in Fig. 11 are for IIR single-layer semi-transparent cirrus clouds having IAB > 0.01 sr-[1] or $\tau$ > ~




0.3. Most of the PSD $\beta_{eff}$ derived using the SPEC relationships are smaller than median IIR $\beta_{eff}$, whereas using the EM2016 relationships brings PSD and IIR $\beta_{eff}$ values in a good agreement. However, because the field campaigns targeted TTL cirrus clouds, most of the PSD temperatures are colder than 208 K whereas IIR sampling is sparse below 200 K. As the sampled region is over oceans, we repeated the experiment in Fig. 12 but this time by including cloud having $\tau > \sim 0.01$. IIR sampling of TTL clouds is improved in Fig. 12, so that the comparisons are more informative. Despite the increased random noise,

which explains the larger occurrence of extreme IIR $\beta_{eff}$ values in Fig. 12 than in Fig. 11, IIR and PSD $\beta_{eff}$ are again in better agreement for the EM2016 relationships. The horizontal dashed light blue lines in the left panels of Figs. 11 and 12 indicate $\beta_{eff}$ at the sensitivity limit (see Sect. 3).

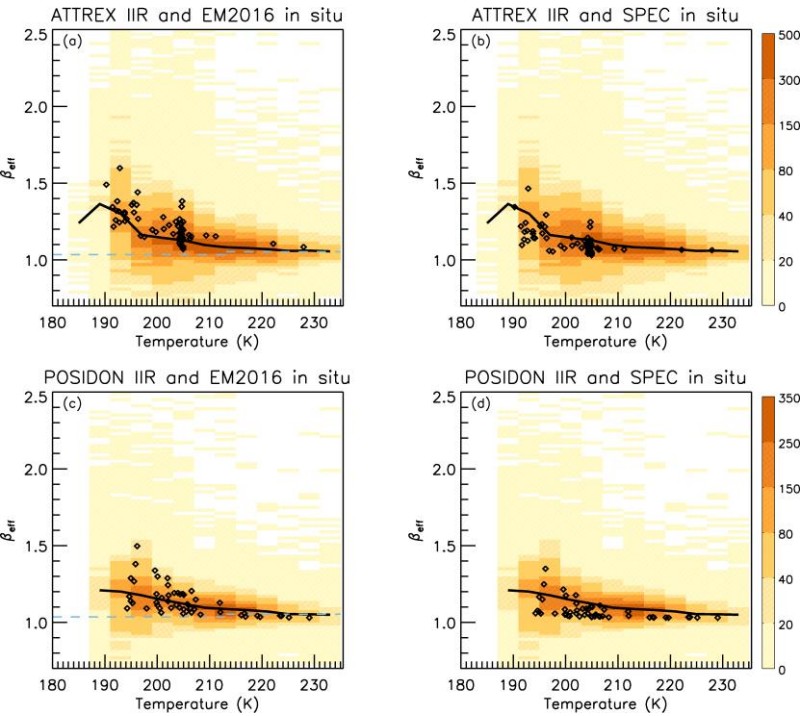

**Figure 12. Same as Fig. 11 but using IIR optical depth > ~ 0.01.**

## 3    Relationships used in the retrieval

### 3.1  Correction of the smallest bin of the 2D-S probes

As discussed in Sect. 2.3, a modification of the smallest bin of the PSDs is needed for the SPARTICUS and TC4 campaigns

where only the 2D-S probe was used. The correction was determined from the analysis of PSDs measured during the ATTREX and POSIDON campaigns since the FCDP was used over the ice particle size-range corresponding to the smallest 2D-S bin.




It is instructive to examine the impact of correcting $N(D)_1$ of the 2D-S probe in terms of the $N_i$ retrieval as described in Eq. (2). The field campaign dependence (and thus the 2D-S probe dependence) of $N_i$ enters through the $\beta_{eff}$ dependent terms in Eq. (2), namely through the $N_i/A_{PSD}$ - $\beta_{eff}$ and the $[1/Q_{abs,eff}\ (12\ \mu m)]$ - $\beta_{eff}$ relationships. From Eq. (2), the product of these two ratios is $N_i/\alpha_{abs}$ which is plotted in Fig. 13, showing the impact of the $N(D)_1$ assumption on the $N_i$ retrieval. There are three assumptions: (1) $N(D)_1$ is unmodified, meaning the $N(D)_1$ measurement is correct, (2) $N(D)_1$ is modified, divided by 10.4 as discussed in Sect. 2.3, and (3) $N(D)_1 = 0$. These three assumptions were evaluated in Fig. 13 using 2D-S PSD data from the SPARTICUS field campaign measured at temperatures less than -38°C. Taking the modified assumption (in navy blue) to be most realistic, it is seen that either assumption (1) or (3) can produce significant errors. Moreover, this reveals the $N_i$ retrieval sensitivity to the size bin for the smallest ice particles. It was fortuitous that both the FCDP and 2D-S probes were flown during the ATTREX and POSIDON field campaigns, which enabled the estimation of a correction factor. Hence forwards only the modified assumption is used for $N(D)_1$ for both SPARTICUS and TC4.

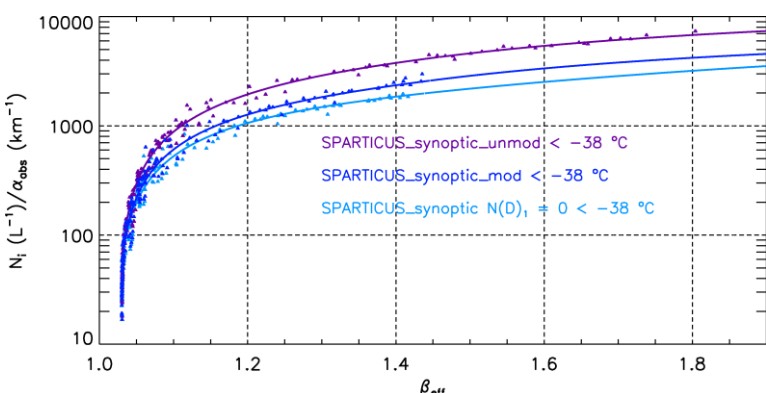

**Figure 13. Sensitivity of the SPARTICUS $N_i$ retrieval to assumptions concerning the first size-bin of the 2D-S probe, $N(D)_1$, which can either be unmodified (purple), modified (by dividing $N(D)_1$ by 10.4, navy blue), or set equal to zero (light blue). See text for details.**

### 3.2 Relating $\beta_{eff}$ to $N_i/IWC$, $N_i/A_{PSD}$, $D_e$, and $1/Q_{abs}(12\ \mu m)$

The X – $\beta_{eff}$ relationships used in the retrieval Eqs. (2), (4), and (7) where X is $N_i/A_{PSD}$, $N_i/IWC$, and $1/Q_{abs}(12\ \mu m)$, respectively, are shown in Fig. 14 along with the $\beta_{eff}$ dependence of $D_e$ which is based on Eq. (4). The solid lines in panels (a), (b), and (d) are second order polynomial curve fits based on the indicated field campaigns where both X and $\beta_{eff}$ are calculated from PSD measurements and MADA. The solid lines in panel (c) are based on Eq. (4). PSDs from the ATTREX and POSIDON field campaigns are mostly from TTL cirrus and were sampled using the same instruments in the tropical western Pacific; therefore, they were combined as a single dataset. While the SPARTICUS data was subdivided into anvil cirrus and synoptic cirrus (i.e., any cirrus not associated with convection), only the curve fits for synoptic cirrus were used





since they represented both cirrus types well. All the PSDs used to produce Fig. 14 were measured at temperatures less than -38°C. The IWCs and $\beta_{eff}$ values were calculated using the m-D expressions in EM2016.

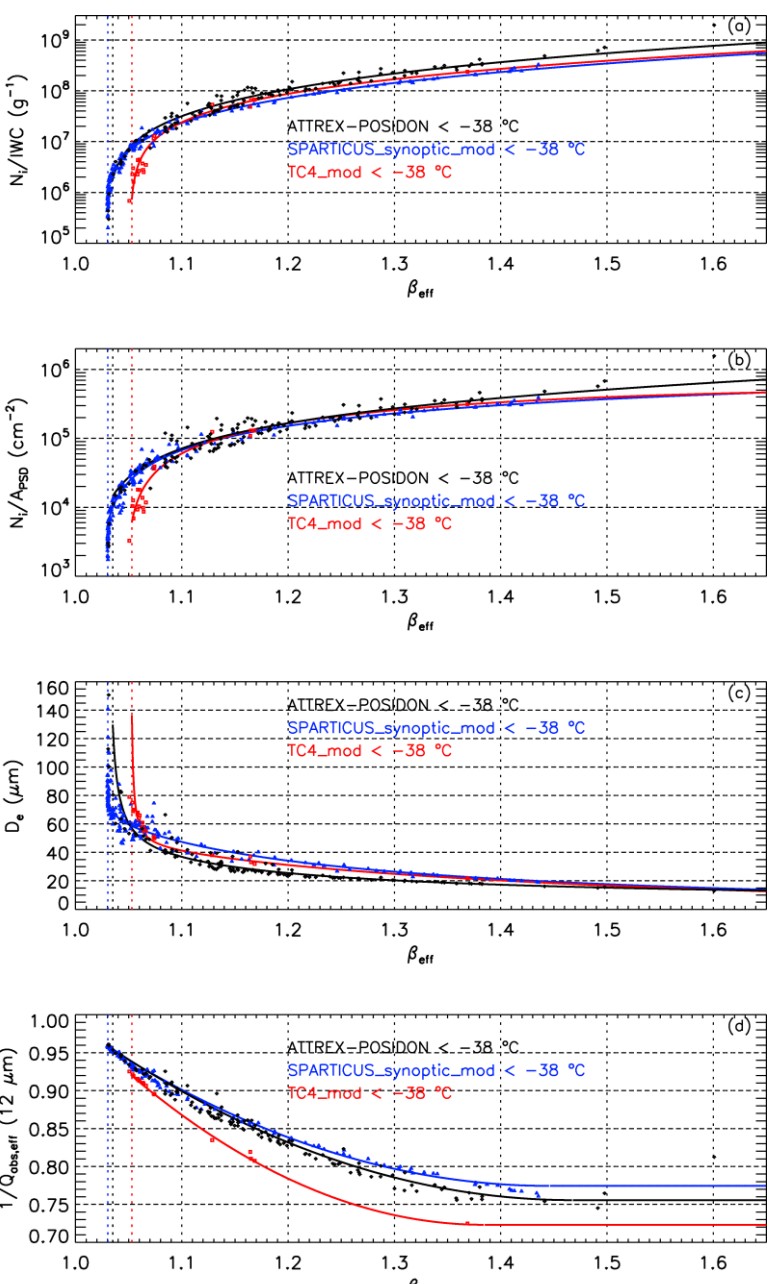


**Figure 14. The dependence of this retrieval on $\beta_{eff}$ is comprised of the above four types of relationships. The curve fits shown correspond to the ATTREX-POSIDON (black), SPARTICUS (navy blue) and TC4 (red) field campaigns where SPARTICUS is based on synoptic cirrus clouds and N(D)₁ was modified for SPARTICUS and TC4. Data points were calculated from the PSD samples having temperature less than -38 °C with colors indicating the respective field campaign.**





Table 3. Regression curve variables and coefficients for polynomials of the form $X = a_0 + a_1 x + a_2 x^2$ used in the CALIPSO retrieval. Units for $N_i$/IWC and $N$/$A_{PSD}$ are in $g^{-1}$ and $cm^{-2}$, respectively.

| X | x | $a_o$ | $a_1$ | $a_2$ |
|---|---|---|---|---|
| SPARTICUS | | | | |
| $N_i$/IWC | $\beta_{eff}$ | $0.84597\times10^9$ | $-1.88517\times10^9$ | $1.03391\times10^9$ |
| $N_i$/$A_{PSD}$ | $\beta_{eff} \leq 2.1$ | $-1.21251 \times 10^6$ | $1.459 \times 10^6$ | $-0.268493 \times 10^6$ |
| | $\beta_{eff} > 2.1$ | $-0.28446 \times 10^5$ | $3.3133 \times 10^5$ | $0$ |
| $1/Q_{abs,eff}$ | $\beta_{eff} \leq 1.45$ | $2.99$ | $-3.065$ | $1.06$ |
| (12 μm) | $\beta_{eff} > 1.45$ | $0.774$ | $0$ | $0$ |
| TC4 | | | | |
| $N_i$/IWC | $\beta_{eff}$ | $0.566052\times10^9$ | $-1.52366\times10^9$ | $0.93712\times10^9$ |
| $N_i$/$A_{PSD}$ | $\beta_{eff} \leq 1.65$ | $-2.03022\times10^6$ | $2.67666\times10^6$ | $-0.705499\times10^6$ |
| | $\beta_{eff} > 1.65$ | $-1.09499 \times 10^5$ | $3.48513\times10^5$ | $0$ |
| $1/Q_{abs,eff}$ | $\beta_{eff} \leq 1.38$ | $4.15$ | $-4.95$ | $1.7875$ |
| (12 μm) | $\beta_{eff} > 1.38$ | $0.723$ | $0$ | $0$ |
| ATTREX-POSIDON | | | | |
| $N_i$/IWC | $\beta_{eff}$ | $1.56577\times10^9$ | $-3.36428\times10^9$ | $1.79055\times10^9$ |
| $N_i$/$A_{PSD}$ | $\beta_{eff}$ | $-0.3480\times10^6$ | $-0.1437\times10^6$ | $0.4772\times10^6$ |
| $1/Q_{abs,eff}$ | $\beta_{eff} \leq 1.47$ | $3.045$ | $-3.12$ | $1.063$ |
| (12 μm) | $\beta_{eff} > 1.47$ | $0.755$ | $0$ | $0$ |

A plot similar to Fig. 14 but showing the curve fits only is included in the Supplement to this article (Fig. S1) and the coefficients used to produce these curves are reported in Table 3, where $\beta_{eff}$ is the independent "x" variable and the retrieved microphysical ratio is the dependent "X" variable. Sometimes a linear extrapolation had to be defined to extend the validity of the formulation over the full range of $\beta_{eff}$. These X – $\beta_{eff}$ relationships in Table 3 are only valid when $\beta_{eff} < 10$ (and are evaluated at $\beta_{eff} = 10$ if $\beta_{eff} > 10$). In practice, $\beta_{eff}$ almost never exceeds 10 and rarely exceeds 2.

As noted in M2018, our retrieval of $N_i$ and $D_e$ is the most sensitive to $\beta_{eff}$ when the PSD includes a large proportion of small ice crystals and $\beta_{eff}$ is relatively large. The vertical dashed lines in Fig. 14 indicate the $\beta_{eff}$ sensitivity limit for each field campaign dataset, which are reported in Table 4. If the retrieved $\beta_{eff}$ lies below this value, the retrieved quantity in the X-column of Table 3 is evaluated at the $\beta_{eff}$ sensitivity limit. Since $D_e$ is essentially a product of two of these ratios, it is constant at the sensitivity limit, as shown in Table 4. However, when the retrieved property has an additional dependence on $\alpha_{ext}$ and therefore $\tau_{abs}$(12.05 μm), that property is not a constant at the sensitivity limit since $\tau_{abs}$(12.05 μm) is not subject to this limit. This is illustrated in Table 4, where the $N_i$ retrieval equation is expressed in terms of the extinction coefficient for visible light





$\alpha_{ext}$. Two values of $\alpha_{ext}$ are given that bracket the $\alpha_{ext}$ range commonly found in cirrus, and corresponding $N_i$ values are given for each $\alpha_{ext}$ and $\beta_{eff}$ sensitivity limit, where $N_i/A_{PSD}$ is evaluated at the sensitivity limit for each campaign.

Table 4: Maximum retrieved $D_e$ (μm) and minimum retrieved $N_i$ (L$^{-1}$) at $\beta_{eff}$ sensitivity limit.

| | $\beta_{eff}$ sensitivity limit | $D_e$ (μm) | $N_i(L^{-1}) = 0.01\dfrac{N_i}{A_{PSD}}(cm^{-2})\dfrac{\alpha_{ext}(km^{-1})}{2}$ | |
|---|---|---|---|---|
| | | | $\alpha_{ext}$ = 0.01 km$^{-1}$ | $\alpha_{ext}$ = 1 km$^{-1}$ |
| SPARTICUS | 1.0304 | 77.7 | 0.29 | 29 |
| TC4 | 1.053 | 136.2 | 0.60 | 60 |
| ATTREX-POSIDON | 1.035 | 129.8 | 0.72 | 72 |

## 3.3  Strategy for a global retrieval scheme

The X- $\beta_{eff}$ relationships from the 3 field campaigns shown in Fig. 14 are overall consistent, but they exhibit differences. The $D_e - \beta_{eff}$ relationship for TC4 differs significantly from that of the ATTREX-POSIDON campaigns, even though these three campaigns were conducted in the tropics, and they both differ from the SPARTICUS relationship obtained at mid-latitudes. For a given $\beta_{eff}$ larger than 1.05, ATTREX-POSIDON yields the smallest $D_e$ and SPARTICUS the largest one. By expressing

PSDs in terms of $d_e$ (effective photon path) as described in Sect. 2.4.1, Fig. 15 shows that TTL PSDs differ substantially from SPARTICUS PSDs over a narrow range of $\beta_{eff}$ (i.e., $\beta_{eff}$ is approximately constant). Since $\beta_{eff}$ is essentially the ratio of two absorption coefficients involving the integration of PSD area, integrals of PSD area are shown. Each panel in Fig. 15 shows a SPARTICUS PSD and a PSD taken from either the ATTREX or POSIDON campaign, having similar $\beta_{eff}$ values. In the bottom are the corresponding $D_e$ values. It is seen that over a very narrow range of $\beta_{eff}$, $D_e$ changes considerably (along with

PSD shape), suggesting that the $D_e - \beta_{eff}$ relationship is subject to changes in PSD shape. The number of TC4 PSDs were much less than for SPARTICUS, precluding the pairing of PSDs of similar $\beta_{eff}$. Nonetheless, it appears likely that PSD shape differences may be responsible for the different $D_e - \beta_{eff}$ relationships regarding the anvil cirrus sampled during TC4 and the TTL cirrus sampled during ATTREX-POSIDON.

Supporting evidence relating to differences between anvil and TTL cirrus is found in Gasparini et al. (2018), which

contrasted in situ cirrus dominating at T < -55°C (including TTL cirrus) with liquid origin cirrus dominating when -55°C < T < -40°C, where the latter are either anvil cirrus formed from deep convection or are glaciated mixed phase clouds. Consistent with Gasparini et al. (2018), Heymsfield et al. (2014) found a $D_e$ discontinuity in cirrus clouds in the tropics and at the top of mid-latitude clouds between ~ -60°C and -65°C, with much smaller $D_e$ at these colder temperatures (their Fig. 11).



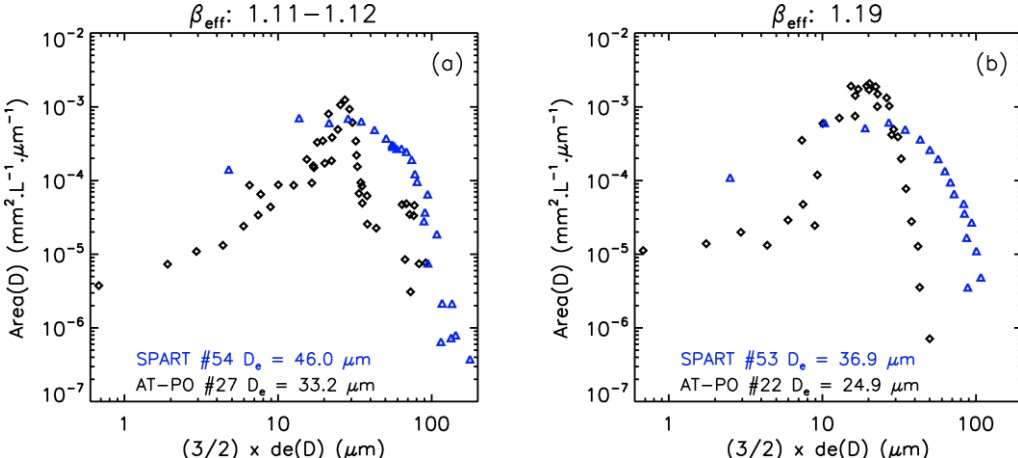

**Figure 15. Comparisons of area PSDs from the ATTREX-POSIDON (black) and SPARTICUS (blue) campaigns having very similar $\beta_{eff}$ values but considerably different $D_e$ values, illustrating how differences in PSD shape between the two field campaigns can yield different $D_e - \beta_{eff}$ relationships.**

Table 5: Combination of the empirical relationships from the various campaigns. The $T_{cold}$ and $T_{warm}$ temperature limits were chosen based on the sampled temperatures during the respective campaigns.

| Cloud temperature, $T_r$ | $T_r \leq T_{cold} = -65$ °C | $T_r \geq T_{warm} = -60$ °C | -65 °C < $T_r$ < -60 °C |
|---|---|---|---|
| Tropics: 30°S - 30°N | ATTREX-POSIDON Cold TTL cirrus, narrow PSD | TC4 | Temperature interpolation for each relationship: $N_i/A_{PSD}$-$\beta_{eff}$, $N_i/IWC$-$\beta_{eff}$, and $1/Q_{abs,eff}(12\,\mu m)$-$\beta_{eff}$ |
| Extra-tropics | ATTREX-POSIDON Narrow PSD | SPARTICUS | |

To accommodate these findings, we developed a latitude- and temperature-dependent scheme for our retrieval as described in Table 5. That is, for clouds having radiative temperature $T_r < -65$°C, the ATTREX-POSIDON $D_e - \beta_{eff}$ relationship was used at any latitude. When $T_r > -60$°C, the TC4 $D_e - \beta_{eff}$ relationship was used in the tropics (30°S - 30°N) and the SPARTICUS $D_e - \beta_{eff}$ relationship was used outside the tropics. Between -60°C and -65°C, a temperature interpolation between the two relevant formulations was implemented. The same practice applies to the other $X - \beta_{eff}$ relationships as shown in Table 3. This temperature dependence mostly affects the tropics as shown in Fig. 16, featuring seasonal maps of the fraction of IIR pixels with cirrus clouds having $T_r < -65$°C relative to all pixels with cirrus clouds (where cirrus clouds are defined as having $T_r \leq 235$ K). These fractions are for $0.01 < \tau < \sim 3$ sampled only over oceans. This fraction was evaluated over both land and ocean using $\sim 0.3 < \tau < \sim 3$ in the Supplement (Fig. S2) where it is shown that in the tropics, the fraction over land is comparable to that over the tropical western Pacific. Over the tropics, this fraction can easily exceed 60% or 70%, while outside the tropics,



this fraction is generally < 5%, with exceptions over the Antarctic (JJA and SON) and over Greenland (DJF) as shown in the Supplement.

The X-$\beta_{eff}$ relationships reported in Table 3 together with the combination strategy summarized in Table 5 can be used to reproduce the findings shown in Sect. 4 and in Part 2 of this study.

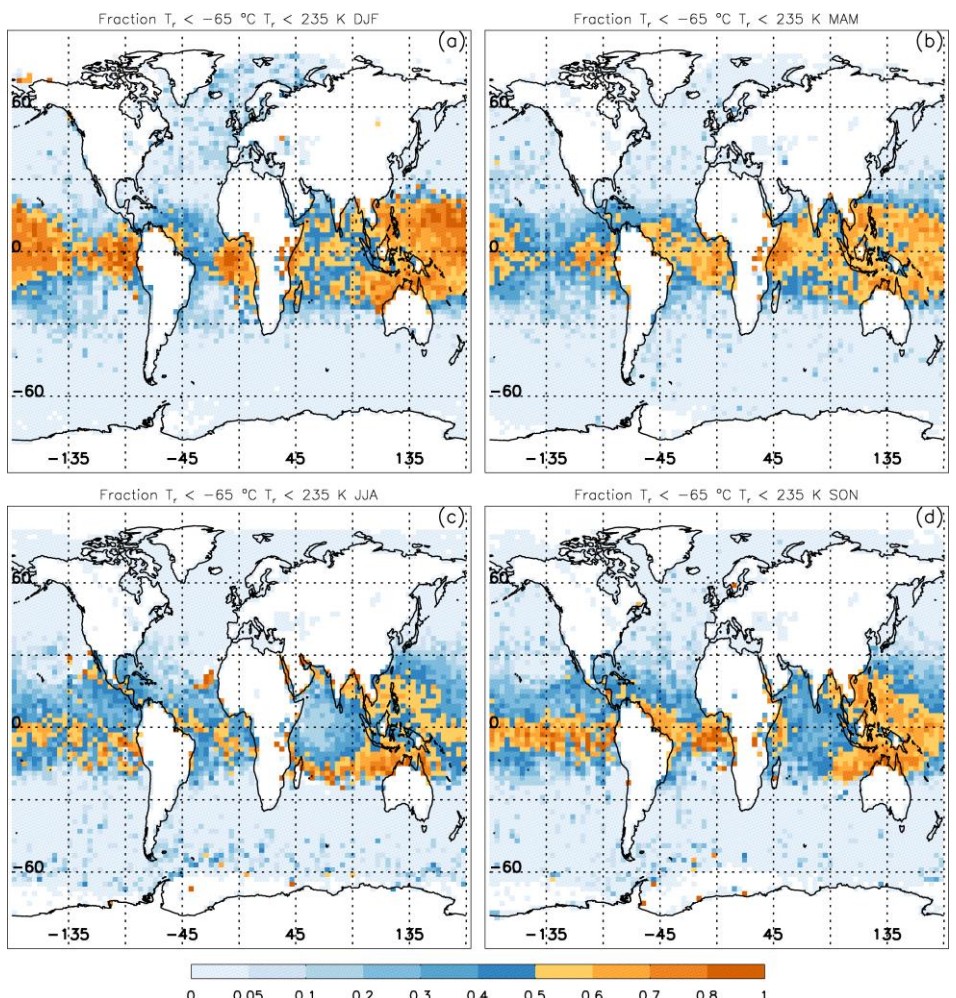

**Figure 16. Seasonal maps of the fraction of cirrus cloud pixels (T$_r$ ≤ 235 K) having T$_r$ < -65°C (208 K) over oceans only, where ~**
**0.01 < τ < ~ 3. This is the fraction of cirrus clouds for which the ATTREX-POSIDON formulation is used in this retrieval. The four panels are for (a) December-January-February (DJF), (b) March-April-May (MAM), (c) June-July-August (JJA), and (d) September-October-November (SON) during 2008, 2010, 2012 and 2013.**



### 3.4 Retrieval uncertainties

Uncertainties in retrieved $\beta_{eff}$, noted $\Delta\beta_{eff}$, translate into uncertainties in $N_i/A_{PSD}$, $N_i/IWC$, $1/Q_{abs,eff}$ (12 µm), and ultimately $D_e$. While $\Delta\beta_{eff}$ increases as optical depth decreases, the resulting uncertainty in X, noted $\Delta X$, depends also on the $\partial X/\partial\beta_{eff}$ slope of the X- $\beta_{eff}$ relationships at retrieved $\beta_{eff}$. An additional contribution to the uncertainty in $N_i$, IWC, and $\alpha_{ext}$ is the uncertainty in $\tau_{abs}$ (12 µm). The uncertainties in $\tau_{abs}$ (12 µm) and $\beta_{eff}$ are estimated following the same rationale as in M2018. Details are given in Appendix B which includes the equations used to estimate the uncertainties in $N_i$, $D_e$, IWC, $\alpha_{ext}$, and $R_v$.

We will see in the following section that relative uncertainties in $N_i$ typically exceed 100 % when $0.01 < \tau < \sim 3$. These large random uncertainties of individual retrievals can be mitigated by accumulating many samples. Median values of an ensemble of retrievals should not be too impacted by the samples having $\beta_{eff}$ smaller than the sensitivity limit for which $N_i/A_{PSD}$, $N_i/IWC$, $1/Q_{abs,eff}$ (12 µm), and ultimately $D_e$ are set to constant values.

### 4       Testing of the retrieval

Since this CALIPSO-IIR retrieval was developed from cirrus cloud field campaign measurements, we compared the satellite retrievals of $N_i$, $D_e$, IWC, and $\alpha_{ext}$ during the period of the field campaigns over their respective regions with these same properties that were measured in situ during the field campaigns. In this section, the retrieval is tested against aircraft measurements from the ATTREX and POSIDON field campaigns for the tropics (together ATPO) and against SPARTICUS aircraft measurements for the midlatitudes. In addition, the Krämer et al. (2020) global climatology of cirrus cloud properties, based on numerous cirrus cloud field campaigns, will be compared against the corresponding properties from this retrieval.


### 4.1       Comparisons with ATTREX and POSIDON PSD data

Since cirrus clouds sampled during these field campaigns were over ocean, their aircraft-measured properties can be compared against corresponding retrieved properties for cirrus having $\sim 0.01 < \tau < \sim 3$, as shown for ATTREX in Fig. 17 and for POSIDON in Fig. 18. These retrievals were confined to the field campaign domain (in the tropical western Pacific) and to the

campaign sampling period (Feb.-March 2014 for ATTREX and Oct. 2016 for POSIDON). The retrieval sample density is given by the color bar while the black diamonds indicate the aircraft in situ PSD measurements for a given property. The blue-dashed curves in the $D_e$ plots indicates the fraction of cirrus clouds for which $D_e$ could be "reliably" retrieved (i.e., $\beta_{eff} > \beta_{eff}$ sensitivity limit). $D_e$ retrievals for which $\beta_{eff} < \beta_{eff}$ sensitivity limit comprise the high sample densities between 130 µm and 136 µm. As PSDs broaden at higher temperatures, $D_e$ increases and $\beta_{eff} < \beta_{eff}$ sensitivity limit occurs more often, which is

evident from POSIDON in situ $D_e$ and the blue-dashed curve. Overall, the ATTREX and POSIDON retrievals appear consistent with the corresponding in situ values. Similar comparisons for optically thicker cirrus where $\sim 0.3 < \tau < \sim 3$ are given in the Supplement (Figs. S3 and S4). Table 6 shows median retrieved properties and relative uncertainty estimates for ATTREX for cirrus having $\sim 0.2 - 0.3 < \tau < \sim 3$ (left) and $\sim 0.01 < \tau < \sim 3$ (right). A similar table for the POSIDON campaign is shown in the Supplement (Table S1). In Table 6, median $\Delta\beta_{eff}$ ranges from 0.03 to 0.44 where median $\tau = 0.05$ at $T_r = 193$



 K, $\Delta N_i/N_i$ = 1.88 and $\Delta D_e/D_e$ = 0.98. The smallest median $\Delta N_i/N_i$ is 0.35 at $T_r$ = 193 K when only the thicker clouds are

sampled and median $\tau$ is somewhat small (0.23), but this is compensated by the fact that $\beta_{eff}$ = 1.57 where the sensitivity of the

technique is very favorable. In contrast, median $\beta_{eff}$ is 1.056-1.058 at 233 K, where the sensitivity of the technique is less

favorable, which explains the occurrence of relative uncertainties larger than 2.4 despite the small $\Delta\beta_{eff}$ = 0.03.

Again, these uncertainty estimates characterize random uncertainties of individual retrievals and are reduced for statistical

 analyses involving a large number of samples.

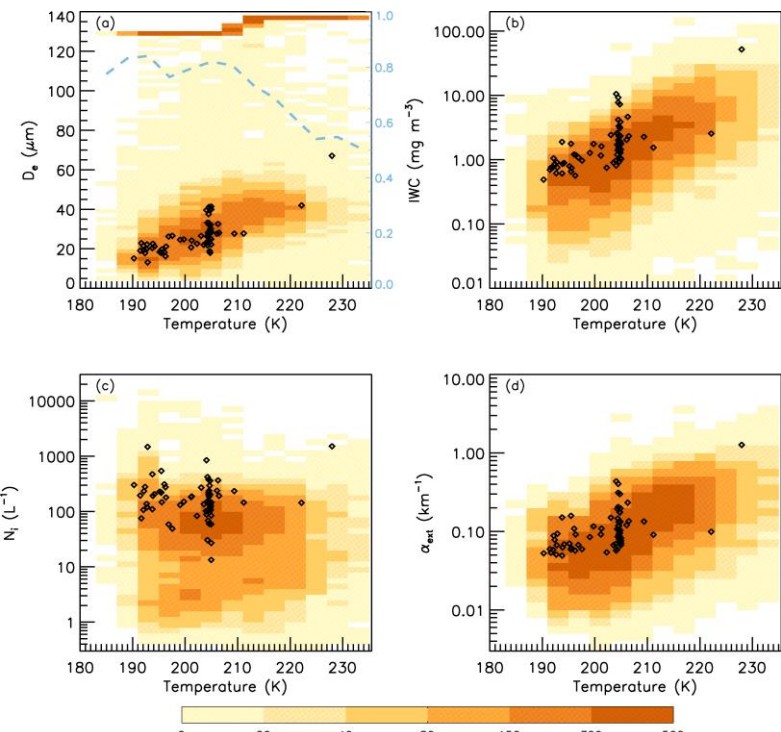

**Figure 17. Pixel sampling densities (given by the color bar) for retrievals of (a) $D_e$, (b) IWC, (c) $N_i$, and (d) $\alpha_{ext}$ taken during the period of the ATTREX field campaign (February-March 2014) over the ATTREX domain (0° N - 20° N and 130° E - 160° E) where**
 **~ 0.01 < $\tau$ < ~ 3. Black diamonds indicate corresponding aircraft PSD measurements of these properties. The right vertical axis of the $D_e$ plot indicates the fraction of cirrus clouds sampled having $\beta_{eff}$ greater than the $\beta_{eff}$ sensitivity limit given by the blue-dashed curve while the high sample densities having $D_e$ between 130 μm and 136 μm are from samples having $\beta_{eff}$ lower than the $\beta_{eff}$ sensitivity limit (i.e., non-quantifiable $D_e$). The change in $D_e$ from 130 to 136 is due to the temperature interpolation (ATPO to TC4).**





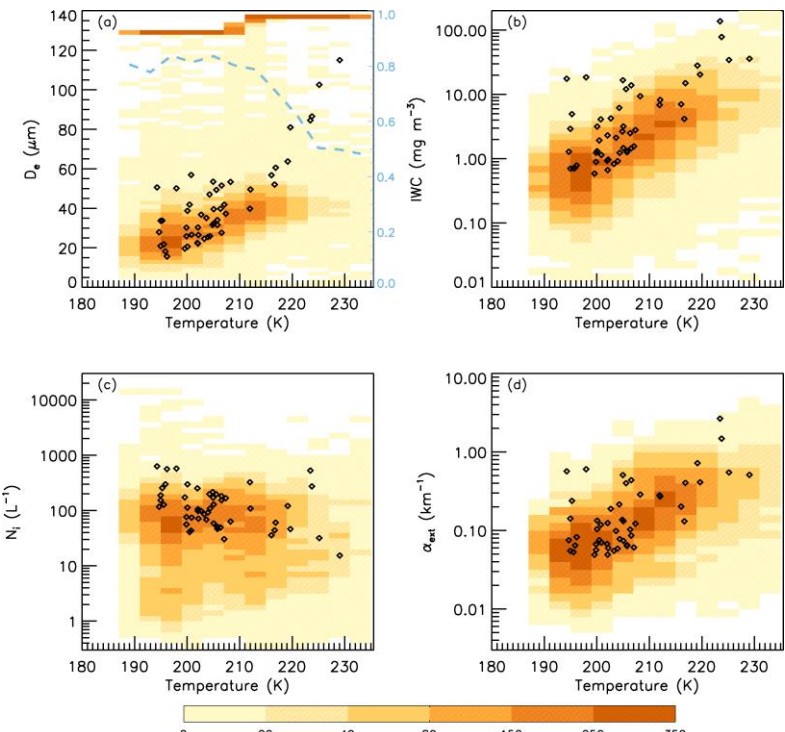

**Figure 18. Same as Fig. 17 but for the POSIDON field campaign in October 2016.**

Table 6: Median values and estimated uncertainties of various retrieved properties at 193, 213, and 233 K for ~ 0.2 - 0.3 < τ < ~ 3 (left) and ~ 0.01 < τ < ~ 3 (right) during the ATTREX campaign.

| **ATTREX** | ~ 0.2 - 0.3 < τ < ~ 3 | | | ~ 0.01 < τ < ~ 3 | | |
|---|---|---|---|---|---|---|
| $T_r$ (K) | 193 | 213 | 233 | 193 | 213 | 233 |
| Pixel count | 144 | 2011 | 253 | 3065 | 3654 | 467 |
| $\tau_{abs}$(12.05 μm) | 0.14 | 0.32 | 0.53 | 0.03 | 0.18 | 0.24 |
| $\Delta\tau_{abs}$(12.05 μm) | 0.016 | 0.019 | 0.028 | 0.016 | 0.019 | 0.023 |
| $\tau$ | 0.23 | 0.56 | 0.95 | 0.05 | 0.32 | 0.42 |
| $\Delta\tau$ | 0.025 | 0.040 | 0.051 | 0.029 | 0.039 | 0.046 |
| $\beta_{eff}$ | 1.569 | 1.083 | 1.058 | 1.301 | 1.083 | 1.056 |
| $\Delta\beta_{eff}$ | 0.18 | 0.03 | 0.03 | 0.44 | 0.05 | 0.05 |
| $\alpha_{vis}$ (km$^{-1}$) | 0.21 | 0.25 | 0.55 | 0.05 | 0.17 | 0.30 |
| $\Delta\alpha_{vis}/\alpha_{vis}$ | 0.13 | 0.07 | 0.05 | 0.63 | 0.12 | 0.11 |
| $D_e$ (μm) | 14 | 44 | 64 | 20 | 44 | 78 |
| $\Delta D_e/D_e$ | 0.19 | 0.19 | 2.44 | 0.98 | 0.45 | > 3.00 |
| IWC (mg m$^{-3}$) | 0.9 | 4.2 | 12.7 | 0.4 | 2.9 | 6.7 |
| $\Delta$IWC/IWC | 0.29 | 0.26 | 2.48 | 1.54 | 0.61 | > 3.00 |
| $N_i$ (L$^{-1}$) | 664 | 52 | 41 | 71 | 34 | 28 |
| $\Delta N_i/N_i$ (L$^{-1}$) | 0.35 | 0.84 | > 3.00 | 1.88 | 1.51 | > 3.00 |





## 4.2 Comparisons with SPARTICUS PSD data

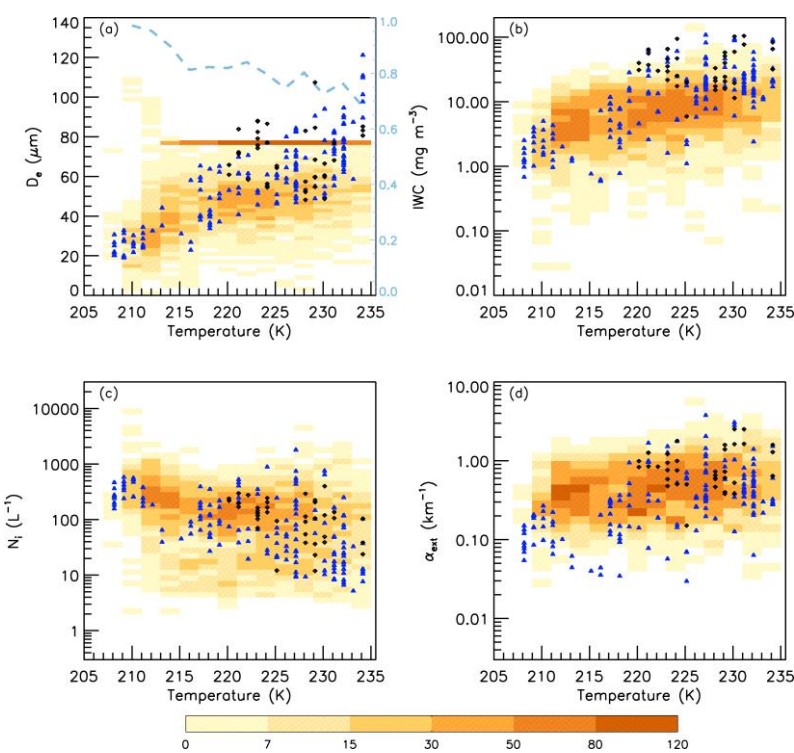

**Figure 19. Pixel sampling densities (given by the color bar) for retrievals of (a) $D_e$, (b) IWC, (c) $N_i$, and (d) $\alpha_{ext}$ taken during the period of the SPARTICUS field campaign (January through April 2010) over the SPARTICUS domain (31- 42 °N; 90-103 °W) where ~ 0.3 < τ < ~ 3. Black diamonds indicate corresponding aircraft PSD measurements of these properties. The right vertical axis of the $D_e$ plot indicates the fraction of cirrus clouds sampled having $\beta_{eff}$ greater than the $\beta_{eff}$ sensitivity limit given by the blue-dashed curve while the high sample densities having $D_e$ ~ 78 μm are from samples having $\beta_{eff}$ lower than the $\beta_{eff}$ sensitivity limit (i.e., non-quantifiable $D_e$).**

As mentioned, $N(D)_1$ of the SPARTICUS PSD data was divided by 10.4 to correct $N(D)_1$ based on a comparison of $N(D)_1$ with corresponding FCDP values from the POSIDON campaign. These corrected SPARTICUS PSDs are used in this section to compare in situ cirrus cloud properties with corresponding retrieved values. As with the ATTREX and POSIDON campaigns, these retrievals are from the SPARTICUS domain (31 – 42 °N and 90 – 103 °W) during the campaign measurement period (January through April 2010). Since these retrievals are over land, they were restricted to the thicker cirrus where ~ 0.3 < τ < ~ 3. These comparisons are shown in Fig. 19. As before, the blue-dashed curve in panel a indicates the fraction of $D_e$ retrievals having $\beta_{eff}$ > the $\beta_{eff}$ sensitivity limit which corresponds to $D_e \approx 78$ μm (shown by the narrow band of high pixel sampling densities). At the highest cirrus temperatures ($T_r$), in situ $D_e$ tends to be higher than retrieved $D_e$ (where $\beta_{eff}$ > the $\beta_{eff}$ sensitivity limit). This may be partly due to aircraft sampling of relatively thick cirrus clouds below the mid-cloud level (i.e., at higher temperatures) where PSDs are broader (with larger $D_e$) due to longer ice particle growth times through vapor diffusion and aggregation. In contrast, retrieved $D_e$ would correspond more closely to mid-cloud temperatures as shown in Fig. 7. Similar reasons may explain why in situ IWCs tend to be higher than retrieved IWCs at higher $T_r$. Overall, the retrievals



in Fig. 19 exhibit reasonable agreement with SPARTICUS in situ measurements, similar to the ATTREX and POSIDON comparisons.

## 4.3  Comparisons with a global cirrus cloud property climatology

A recent study by Krämer et al. (2020) has expanded the in situ cirrus cloud property database described in Krämer et al.
(2009) by a factor of 5 to 10 (depending on cloud property). Here we compare the temperature dependence of $R_v$, $N_i$, and IWC from the CALIPSO-IIR retrievals and from the Krämer et al. (2020) climatology. Since the aircraft measurements used in Krämer et al. (2020) often did not allow $D_e$ to be calculated (and thus $D_e$ was not reported), we use $R_v$ as a measure of ice particle size for comparison purposes since $R_v$ is reported in Krämer et al. (2020). However, $R_v$ and $D_e$ are unique quantities where $D_e$ cannot be calculated from $R_v$ (and vice-versa). Since $D_e$ partly determines a cloud's radiative properties, $D_e$ and $R_v$
are intercompared in Appendix C based on in situ data and for different PSD shape assumptions using a PSD model that assumes a simple gamma PSD distribution. While natural PSDs exhibit shapes more complex than these gamma PSDs, this modelling exercise suggests the relation between $D_e$ and $R_v$ depends on PSD shape.

Figure 3 in Krämer et al. (2020) shows that aircraft measurements are mostly between 20° S and 63° N. Thus, the IIR retrievals were averaged over oceans for 20 °S- 0°, 0°-30° N, and 30° N-63° N for 4 years (2008, 2010, 2012 and 2013). Since
the Krämer et al. (2020) data has no seasonal dependence, IIR retrievals were averaged over all seasons. The results are shown in Fig. 20, where the IIR results, using $T_r$ for the temperature, are in red for samples with $\tau > \sim 0.3$ and in blue using $\tau > \sim 0.01$. In situ data (black curves) in panels a and c are the climatological values. In panel d showing IWC, the black curve is an estimate of median in situ IWC derived from median in situ $R_v$ and median in situ $N_i$ using Eq. (5). The retrieved values of $R_v$, $N_i$, and IWC for $\tau > 0.01$ (blue curves) are generally within the $\pm$ 25 percentile range of corresponding in situ values.

The large spread of IIR data when $\tau$ can be as low as $\sim 0.01$ (blue) compared to $\tau > \sim 0.3$ (red) is due in part to larger random uncertainties in clouds having optical depth $< 0.3$, which represent the majority of the samples at T $< 215$ K (panel b). We note however that median $R_v$ from the red and blue curves are similar, suggesting no systematic bias introduced by the retrievals at $\tau < \sim 0.3$. IIR and in situ median $R_v$ agree reasonably well at T $> 210$ K and below 190 K. IIR $R_v$ increases steadily with temperature and can be lower than in situ $R_v$ by up to 7 μm between 190 and 205 K.

Differences between the optically thicker ($\tau > 0.3$, red) and thinner ($\tau > 0.01$, blue) $N_i$ and IWC retrievals may be due to differences in ice nucleation processes (i.e., het and hom) as described in Part 2, with hom occurring more often in the optically thicker cirrus clouds, promoting higher $N_i$ and IWC. If true, it may be important during cirrus cloud field campaigns to attempt to characterize the cirrus in terms of $\tau$ to make in situ cloud property comparisons with cirrus cloud remote sensing and climate modelling results more meaningful. Fortunately, Krämer et al. (2020) contains a disclaimer stating, "Because of the dangerous
nature of measurements under such conditions, the frequency of convective – and also orographic wave cirrus – is underrepresented in the entire in situ climatology." And related to this, there is a statement about the higher in situ $N_i$ in Krämer et al. (2009) resulting from flights in the "lee wave cirrus behind the Norwegian mountains". Orographic gravity waves (OGWs) produce relatively high updrafts more conducive to hom and tend to produce optically thicker cirrus clouds



with higher $N_i$ that can be spatially extensive (M2018). The sparsity of OGW cirrus in situ sampling in Krämer et al. (2020)
may help explain the tendency of IIR $N_i$ being slightly higher than in situ $N_i$ in Fig. 20.

The retrieved median $N_i$ in Fig. 20 (blue curve) exhibit similar magnitudes as a function of temperature to those of the
DARDAR $N_i$ retrieval (Sourdeval et al., 2018), which are compared against the median $N_i$ of the Krämer et al. climatology in
Fig. 15 of Krämer et al. (2020). The main difference between the DARDAR $N_i$ retrieval and this one is that median DARDAR
$N_i$ is higher for T < 220 K, with DARDAR $N_i$ ~ 100 L$^{-1}$ for T < 205 K.


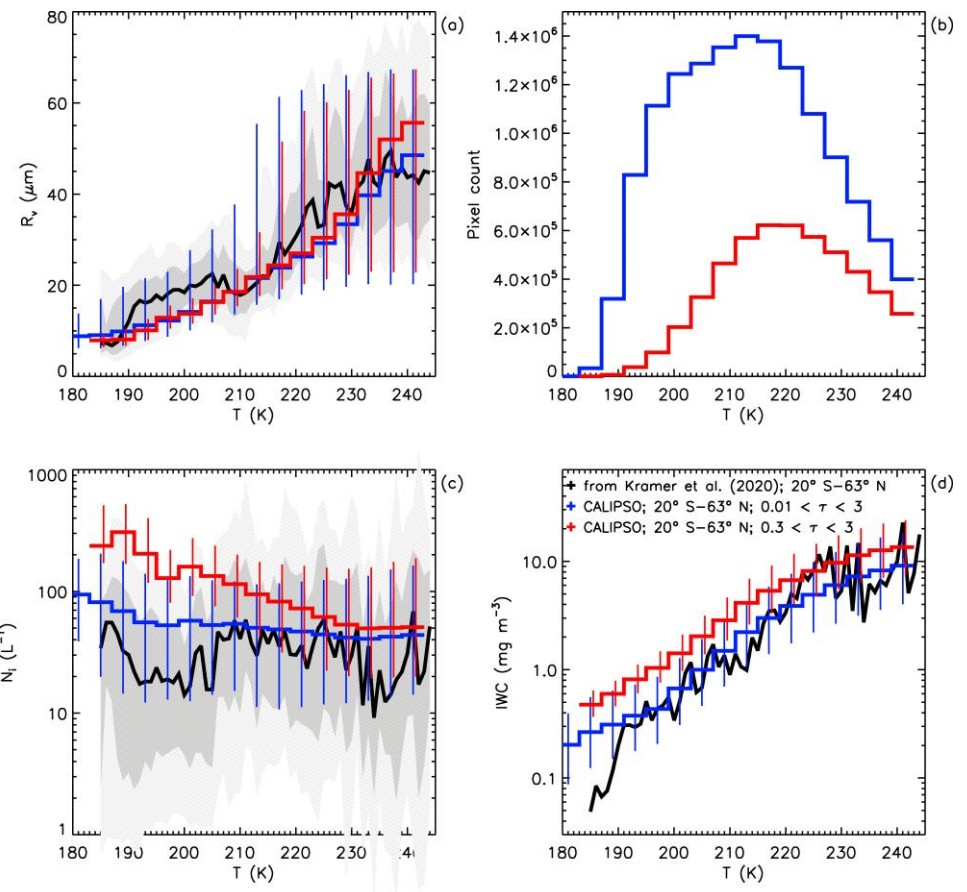

**Figure 20.** Temperature dependence of median values of (a) $R_v$ (µm), (c) $N_i$ (L$^{-1}$), and (d) IWC (mg m$^{-3}$) from the IIR retrievals (red:
~ 0.3 < τ < ~ 3; blue: ~ 0.01 < τ < ~ 3) and from the Krämer et al. (2020) in situ climatology (black curves). The vertical bars indicate
the IIR 25$^{th}$ and 75$^{th}$ percentiles, except in panel (b) which shows the number of IIR sampled pixels. The light shade of grey in panels
(a) and (c) is between the 10$^{th}$ and 90$^{th}$ percentiles and the superimposed darker shade of grey is between the 25$^{th}$ and 75$^{th}$ percentiles
for the in situ data.





# 5    Conclusions

This study has utilized the CALIPSO IIR and CALIOP lidar in new ways, resulting in new methods for retrieving $N_i$, $D_e$, IWC,

IWP, $\alpha_{ext}$, and $\tau$. The following improvements contributed to this CALIPSO retrieval:

1) By expanding the sampling range to include optically thinner cirrus clouds ($0.01 < \tau < 3$), the sampling has become more representative of all cirrus clouds.

2) The retrieval of $N_i$ has become more accurate by using the $N_i/A_{PSD}$ ratio, which is directly measured by aircraft probes.

3) The retrieval of $D_e$ has become more accurate by using the ratios $N_i/A_{PSD}$ and $N_i/IWC$, where IWC is estimated using

mass-dimension relationships that appear more realistic.

4) Improvements in $D_e$ accuracy transfer to improvements in IWC and IWP accuracy via Eqs. 6 and 9, respectively.

5) The relationship between $D_e$ and $\beta_{eff}$ was not unique, where PSDs having the same $\beta_{eff}$ can have different $D_e$ due to PSD shape differences between TTL cirrus and cirrus at higher temperatures. For this reason, separate X - $\beta_{eff}$ relationships were developed for TTL and anvil (or synoptic) cirrus, with a temperature interpolation linking these

two temperature regimes. This mostly affects the tropics where cirrus clouds are abundant in the TTL (see Fig. 16).

6) By comparing the FCDP and 2D-S probes in their overlap region, the first size-bin of the 2D-S probe was corrected to a first approximation, resulting in improved X - $\beta_{eff}$ relationships.

7) In general, the physical properties of cirrus clouds differ when comparing optically thicker ($0.3 < \tau < 3$) cirrus clouds with all cirrus clouds ($0.01 < \tau < 3$), where $N_i$ and IWC are higher in the optically thicker cirrus clouds.

In view of (7), cirrus cloud field campaigns should indicate, if possible, the type of cirrus clouds being sampled, especially outside the tropics where orographic wave (OGW) cloud cirrus (often having $\tau > 0.3$) are common (M2018). A global/seasonal analysis of the frequency of occurrence of these OGW cirrus clouds, developed through satellite remote sensing, would also be useful for testing the representation of cirrus clouds in climate models, given their distinct optical properties.

Given the apparent dependence of $N_i$ and IWC on $\tau$, the agreement between the two remote sensing methods (DARDAR

and CALIPSO) and the Krämer et al. (2020) climatology appears reasonable. That is, cirrus associated with strong updrafts (i.e., anvil cirrus near convection and OGW cirrus) are generally avoided during cirrus field campaigns for safety reasons (Krämer et al., 2020) and therefore may not be accurately represented by in situ sampling-based climatology. It may be possible that the high median DARDAR $N_i$ (~ 100 $L^{-1}$) for T < 205 K (Krämer et al. 2020, Fig. 15) relative to in situ climatological $N_i$ in Fig. 20 results from the DARDAR sampling of thick anvil cirrus near convection where hom affects $N_i$

more profoundly. This CALIPSO retrieval does not sample such cirrus (i.e., $\tau > 3$) and thus would retrieve a lower median climatological $N_i$. Nonetheless, tropical cirrus clouds having $\tau < 3$ are probably representative of tropical cirrus in terms of their areal coverage, which matters most for cloud radiative effects.

The application of this CALIPSO retrieval for studying the physics of cirrus clouds is exemplified in Part 2 of this article. In particular, a method for estimating the fraction of cirrus clouds strongly affected by hom is presented as well as a new

conceptual model for cirrus cloud formation and evolution.



## 6    APPENDIX A: Inter-channel optical depth differences

Both over land and over oceans, the solid lines in Fig.3 and Fig. 4 tend to 0 as IAB tends to 0, as expected. The median $\tau_{abs}$12-10 differences are reported in Table A1. To estimate whether these differences are realistic, Table A1 also includes an approximate $\beta_{eff}$ derived from the median $\tau_{abs}$12-10 and median $\tau_{abs}$(12.05 µm) reported in Table 1 as

$$\beta_{eff}\ proxy\ =\ \frac{median\ \tau_{abs}(12.05\ \mu m)}{median\ \tau_{abs}(12.05\ \mu m)\ -\ median\ \tau_{abs}12-10}\ .$$    (A1)

This approximate $\beta_{eff}$ is the ratio of two very small optical depths (smaller than 0.01) and is therefore very sensitive to small inter-channel biases. It is as expected larger than 1, except at 0° N-30° N over land in DJF where it is only slightly smaller. We estimate that the upper range of realistic values for $\beta_{eff}$ is ~ 1.5-2, so that $\beta_{eff}$ = 3.2 at 60°-82° N over oceans in DJF is unambiguously overestimated. Decreasing $\tau_{abs}$12-10 from 0.0029 to 0.0010 would bring $\beta_{eff}$ to 1.3, suggesting a positive 12-

10 inter-channel bias ≤ 0.002 at 60°-82° N over oceans in DJF. Note that the impact of such an inter-channel bias decreases sharply as optical depth increases (Garnier et al., 2021a). For instance, at $\tau_{abs}$(12.05 µm) = 0.05, corresponding to IAB ~ 0.004 in Fig. 2, $\beta_{eff}$ = 1.2 in Fig. 4 could correspond to true $\beta_{eff}$ ~ 1.145, i.e. $\beta_{eff}$ could be overestimated but less than 0.06. At $\tau_{abs}$(12.05 µm) = 0.15, corresponding to IAB ~ 0.01 in Fig. 2, $\beta_{eff}$ = 1.15 in Fig. 4 could be overestimated by less than 0.02.

Table A1: Median IIR $\tau_{abs}$(12.05 µm) - $\tau_{abs}$(10.6 µm) (i.e., $\tau_{abs}$12-10) at CALIOP IAB ~ 7.6 x $10^{-4}$ $sr^{-1}$ using all retrievals (cf solid lines in Figs. 3 and 4) and an approximation for $\beta_{eff}$ (Eq. (A1)).

| | Land | | | | Oceans | | | |
| --- | --- | --- | --- | --- | --- | --- | --- | --- |
| | DJF | | JJA | | DJF | | JJA | |
| Latitude | $\tau_{abs}$12-10 | $\beta_{eff}$ proxy | $\tau_{abs}$12-10 | $\beta_{eff}$ proxy | $\tau_{abs}$12-10 | $\beta_{eff}$ proxy | $\tau_{abs}$12-10 | $\beta_{eff}$ proxy |
| 60°-82° N | 0.0024 | 1.17 | 0.0010 | 1.10 | 0.0030 | 3.20 | 0.0016 | 1.22 |
| 30°-60° N | 0.0020 | 1.45 | 0.0007 | 1.10 | 0.0018 | 1.39 | 0.0008 | 1.15 |
| 0°-30° N | -0.0002 | 0.97 | 0.0008 | 1.34 | 0.0009 | 1.20 | 0.0006 | 1.09 |
| 30°-0° S | 0.0015 | 1.87 | 0.0015 | 1.48 | 0.0007 | 1.15 | 0.0006 | 1.11 |
| 60°-30° S | 0.0013 | 1.17 | 0.0015 | 1.22 | 0.0012 | 1.16 | 0.0015 | 1.23 |
| 82°-60° S | 0.0011 | 1.06 | 0.0021 | 1.21 | 0.0031 | 1.86 | 0.0027 | 1.90 |

## 7    APPENDIX B: Retrieval uncertainty analysis

### 7.1  Ice particle number concentration, $N_i$

The retrieval equation for the ice particle number concentration, $N_i$, is

$$N_i = \left[\frac{N_i}{A_{PSD}}\right]_{\beta_{eff}} \frac{\tau_{abs}(12.05\ \mu m)}{[Q_{abs,eff}(12\ \mu m)]_{\beta_{eff}} \Delta z_{eq}}\ .$$    (B1)





The quantities $N_i/A_{PSD}$ and $1/Q_{abs,eff}(12 \mu m)$ are retrieved from $\beta_{eff}$ using regression curves and the coefficients given in Table 3. By writing $x = \beta_{eff}$, they are computed as

$$\left(\frac{N_i}{A_{PSD}}\right)(cm^{-2}) = \sum_{i=0}^{i=2} b_i x^i \tag{B2}$$

and

$$\frac{1}{Q_{abs,eff}(12\mu m)} = \sum_{i=0}^{i=2} c_i x^i . \tag{B3}$$

Equation (B1) can be re-written as

$$N_i(L^{-1}) = 0.01 \times f(x)(cm^{-2}) \times \alpha_{abs}(km^{-1}) \tag{B4}$$

with

$$f(x) = \left(\sum_{i=0}^{i=2} b_i x^i\right)\left(\sum_{i=0}^{i=2} c_i x^i\right) \tag{B5}$$

and

$$\alpha_{abs} = \frac{\tau_{abs}(12.05\mu m)}{\Delta z_{eq}} . \tag{B6}$$

Assuming a negligible error in $\Delta Z_{eq}$, and writing $\tau_{abs}(12.05 \mu m)$ as $\tau_{12}$ and $\tau_{abs}(10.6 \mu m)$ as $\tau_{10}$ for more clarity, so that $x = \tau_{12}/\tau_{10}$, the derivative of $N_i$ can be written

$$\frac{dN_i}{N_i} = \frac{1}{f}\frac{\partial f}{\partial x} x \left(\frac{d\tau_{12}}{\tau_{12}} - \frac{d\tau_{10}}{\tau_{10}}\right) + \frac{d\tau_{12}}{\tau_{12}} . \tag{B7}$$

In Eq. (B7), the derivative of $x = \beta_{eff}$ is

$$dx = d\beta_{eff} = x \left(\frac{d\tau_{12}}{\tau_{12}} - \frac{d\tau_{10}}{\tau_{10}}\right) . \tag{B8}$$

Errors in $\tau_{12}$ and in $\tau_{10}$ are computed by propagating errors in i) the measured brightness temperatures $T_m$, ii) the background brightness temperatures $T_{BG}$, and iii) the blackbody brightness temperatures $T_{BB}$ (Garnier et al., 2015; Garnier et al., 2021a, M2018). For each of the 3 temperature (T) components ($T_{BG}$, $T_{BB}$, and $T_m$), $\frac{\partial \tau}{\partial T}$ in channels 12 and 10 is computed as

$$\frac{\partial \tau}{\partial T} = \frac{1}{1-\varepsilon}\frac{\partial \varepsilon}{\partial T} \tag{B9}$$

where the effective emissivity in the channel, $\varepsilon$, and the three associated $\frac{\partial \varepsilon}{\partial T}$ terms are reported in the IIR Version 4 product.

The uncertainties $\Delta T_{m10}$ in $T_{m10}$ at 10.6 $\mu m$ and $\Delta T_{m12}$ in $T_{m12}$ at 12.05 $\mu m$ are random errors set to 0.3 K, which are statistically independent (Garnier et al., 2015; Garnier et al., 2021a). Because the same cloud temperature is used to compute $\tau_{12}$ and $\tau_{10}$, the uncertainty $\Delta T_{BB}$ in $T_{BB}$ is the same at 10.6 and at 12.05 $\mu m$. A random error of +/-2K is estimated to include



errors in the atmospheric model. Finally, it was shown in Garnier et al. (2021a) that the uncertainty $\Delta T_{BG}$ in $T_{BG}$ can be considered identical in both channels. $\Delta T_{BG}$ is estimated to be 1 K over oceans and 3 K over land.

Finally, the relative uncertainty $\Delta N_i/N_i$ is written as

$$\left(\frac{\Delta N_i}{N_i}\right)^2 = \left[\frac{1}{f}\frac{\partial f}{\partial x}x\left(\frac{\partial \tau_{12}}{\tau_{12}\,\partial T_{BG}} - \frac{\partial \tau_{10}}{\tau_{10}\,\partial T_{BG}}\right) + \frac{\partial \tau_{12}}{\tau_{12}\,\partial T_{BG}}\right]^2 \Delta T_{BG}^2 + \left[\frac{1}{f}\frac{\partial f}{\partial x}x\left(\frac{\partial \tau_{12}}{\tau_{12}\,\partial T_{BB}} - \frac{\partial \tau_{10}}{\tau_{10}\,\partial T_{BB}}\right) + \frac{\partial \tau_{12}}{\tau_{12}\,\partial T_{BB}}\right]^2 \Delta T_{BB}^2 +$$

$$\left[\left(\frac{1}{f}\frac{\partial f}{\partial x}x + 1\right)\frac{\partial \tau_{12}}{\tau_{12}\,\partial T_{m12}}\right]^2 \Delta T_{m12}^2 + \left[\left(\frac{1}{f}\frac{\partial f}{\partial x}\,x\right)\frac{\partial \tau_{10}}{\tau_{10}\,\partial T_{m10}}\right]^2 \Delta T_{m10}^2 \,.$$

(B10)

## 7.2 Effective diameter, $D_e$

$$D_e = \frac{3}{2\rho_i}\left[\frac{N_i}{A_{PSD}}\right]_{\beta_{eff}}\left[\frac{IWC}{N_i}\right]_{\beta_{eff}}$$ (B11)

with $\rho_i = 0.917$ g.cm$^{-3}$. Again, $(N_i/A_{PSD})$ is given by Eq. (B2), and $(IWC/N_i)$ is retrieved from $x = \beta_{eff}$ as

$$\frac{IWC}{N_i}(g) = 1/\sum_{i=0}^{i=2} d_i x^i.$$ (B12)

It comes that Eq. (B11) can be re-written

$$D_e(\mu m) = 10^4 \times \left(\frac{3}{2\rho_i}\right) \times g(x)(g\,cm^{-2}) = 10^4\left(\frac{3}{2\rho_i}\right)\frac{\sum_{i=0}^{i=2} b_i x^i}{\sum_{i=0}^{i=2} d_i x^i}.$$ (B13)

Using the same notations as previously, it comes that

$$\frac{dD_e}{D_e} = \frac{1}{g}\frac{\partial g}{\partial x}\,x\left(\frac{d\tau_{12}}{\tau_{12}} - \frac{d\tau_{10}}{\tau_{10}}\right).$$ (B14)

Finally, the relative uncertainty $\Delta D_e/D_e$ is written as

$$\left(\frac{\Delta D_e}{D_e}\right)^2 = \left[\frac{1}{g}\frac{\partial g}{\partial x}x\left(\frac{\partial \tau_{12}}{\tau_{12}\,\partial T_{BG}} - \frac{\partial \tau_{10}}{\tau_{10}\,\partial T_{BG}}\right)\right]^2 \Delta T_{BG}^2 + \left[\frac{1}{g}\frac{\partial g}{\partial x}x\left(\frac{\partial \tau_{12}}{\tau_{12}\,\partial T_{BB}} - \frac{\partial \tau_{10}}{\tau_{10}\,\partial T_{BB}}\right)\right]^2 \Delta T_{BB}^2$$

$$+ \left[\frac{1}{g}\frac{\partial g}{\partial x}\,x\,\frac{\partial \tau_{12}}{\tau_{12}\,\partial T_{m12}}\right]^2 \Delta T_{m12}^2 + \left[\frac{1}{g}\frac{\partial g}{\partial x}x\frac{\partial \tau_{10}}{\tau_{10}\,\partial T_{m10}}\right]^2 \Delta T_{m10}^2.$$ (B15)

## 7.3 Ice water content, IWC

$$IWC = N_i\left[\frac{IWC}{N_i}\right]_{\beta_{eff}}$$ (B16)

Using Eq. (B4), Eq. (B16) can be re-written

$$IWC\,(mg\,m^{-3}) = 10^4 \times h(x)(g\,cm^{-2}) \times \alpha_{abs}(km^{-1})$$ (B17)



with

$$h(x) = f(x) / \sum_{i=0}^{i=2} d_i x^i. \tag{B18}$$

It comes that the relative uncertainty $\Delta IWC/IWC$ is given by Eq. (B10) by replacing $f(x)$ with $h(x)$.

### 7.4 Visible IIR equivalent extinction coefficient, $\alpha_{ext}$

$$\alpha_{ext} = 2 \left[ \frac{1}{Q_{abs,eff}(12\,\mu m)} \right]_{\beta_{eff}} \frac{\tau_{abs}(12.05\,\mu m)}{\Delta z_{eq}} \tag{B19}$$

Equation (B19) can be written

$$\alpha_{ext}(km^{-1}) = k(x) \times \alpha_{abs}(km^{-1}) = 2 \left( \sum_{i=0}^{i=2} c_i x^i \right) \times \alpha_{abs}(km^{-1}) \tag{B20}$$

Again, the relative uncertainty $\Delta\alpha_{ext}/\alpha_{ext}$ is given by Eq. (B10), using now $k(x)$ instead of $f(x)$.

### 7.5 Volume radius, $R_v$

$$R_v = \left( \frac{3}{4\pi\rho_i} \right)^{1/3} \left[ \frac{IWC}{N_i} \right]_{\beta_{eff}}^{1/3} \tag{B21}$$

Which can be written as

$$R_v(\mu m) = 10^4 \times \left( \frac{3}{4\pi\rho_i} \right)^{1/3} \times l(x) = 10^4 \left( \frac{3}{4\pi\rho_i} \right)^{1/3} \left( \sum_{i=0}^{i=2} d_i x^i \right)^{-1/3}. \tag{B22}$$

Eq. (B22) is of the same form as Eq. (B13) and the relative uncertainty $\Delta R_v/R_v$ is given by Eq.(B15) by replacing $g(x)$ with $l(x)$.

### 795  8  APPENDIX C: Relating $R_v$ to $D_e$

Figure C1 shows the relationship between $D_e$ and $R_v$ for the PSDs measured at temperatures colder than -38°C during the ATTREX-POSIDON, SPARTICUS (synoptic cirrus clouds only) and TC4 field campaigns. We recall that PSDs measured during SPARTICUS and TC4 were modified by dividing $N(D)_1$ measured in the first bin by a correction factor equal to 10.4 (see Sect. 2.3). For reference, the curves in grey show relationships assuming a simple gamma PSD distribution expressed as

$$N(D) = N_0 D^\upsilon e^{-\lambda D}, \tag{C1}$$

where D is the ice particle maximum dimension, $\upsilon$ is the PSD dispersion parameter and $\lambda$ is the PSD slope in log $N(D)$ – D space and where $\upsilon$ is varied between -0.5 and 4.0. $R_v$ and $D_e$ of a PSD were computed using mass-dimension and area-dimension relationships from EM2016 for anvil cirrus clouds between -55 and -40 °C in combination with a temperature-dependent PSD scheme for tropical anvil cirrus clouds (Mitchell et al., 1999) where only the large ice particle mode was used.



They are independent of the multiplying term, $N_0$. The small-particle end of the PSD is governed by ν, with decreasing

contributions from these smaller particles as ν increases. The simulated PSDs reproduce the general behaviour seen in the in

situ data, and they illustrate the dependence of the $D_e$-$R_v$ relationship on the PSD shape. The exponential form of Eq. (C1)

(i.e., ν = 0) gives an approximate representation for those cirrus clouds having relatively high concentrations of small ice

crystals, assuming that the mid-to-large sizes follow an exponential distribution and $D_e$ is > 50 μm. For narrow PSDs having

$D_e$ < 50 μm, ν tends to be >> 0.

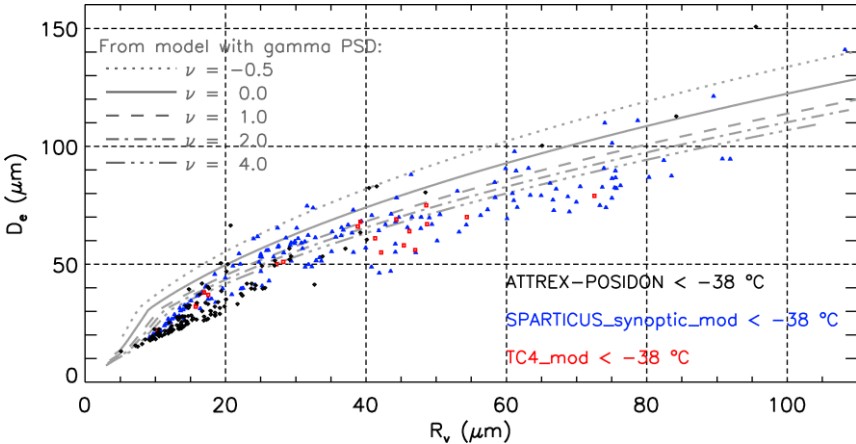

**Figure C1. $D_e$ against $R_v$ for the PSDs measured at T < -38° C during the ATTREX-POSIDON (black), SPARTICUS (navy blue) and TC4 (red) field campaigns where SPARTICUS is based on synoptic cirrus clouds and N(D)₁ was modified for SPARTICUS and**
**TC4. The grey curves are from a model with gamma PSD for 5 values of the PSD dispersion parameter, ν, between -0.5 and 4, illustrating that PSD shape affects the $D_e$-$R_v$ relationship.**

## 9    Data availability

The CALIPSO IIR Level 2 data products used in this study are available at the NASA Langley Atmospheric Science Data

Center and can be retrieved from https://doi.org/10.5067/IIR/CALIPSO/CAL_IIR_L2_Track-Standard-V4-51. The CALIPSO

Lidar Level 2 cloud profiles used in this study are available at the NASA Langley Atmospheric Science Data Center and can

be retrieved from https://doi.org/10.5067/CALIOP/CALIPSO/CAL_LID_L2_05kmCLay-Standard-V4-51. These CALIPSO

data products are also available from the AERIS/ICARE Data and Services Center in France (https://www.icare.univ-lille.fr/).

SPARTICUS in situ data is available from the ARM Data Archive at https://www.arm.gov/data/. ATTREX, POSIDON, and

TC4 in situ data is available at https://espoarchive.nasa.gov/archive/browse.





## 10    Author contribution

DM and AG conceived the study; DM analysed the in situ aircraft data and contributed to the writing of the paper; AG accessed and analysed the CALIPSO data and contributed to the analysis of the in situ aircraft data and to the writing of the paper. SW accessed and processed the ATTREX and POSIDON aircraft data.

## 11    Competing interests

The authors declare that they have no conflict of interest.

## 12    Acknowledgements.

This research was supported by the NASA CALIPSO project and by NOAA grant NA22OAR4690640. Dr. Paul Lawson is gratefully acknowledged for his assistance in providing the aircraft in situ data for the ATTREX and POSIDON field campaigns. We are also grateful to Prof. Martina Krämer for providing the global climatology of cirrus cloud properties. The authors are grateful to the Centre National d'Etudes Spatiales (CNES) and to the AERIS/ICARE Data and Services Center in France for their support with the CALIPSO IIR data.

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
