# Peer review of "Advances in CALIPSO (IIR) cirrus cloud property retrievals – Part 1: Methods and testing"

_EGUsphere, 2024_

## Referee Comment (RC2)

Review of "Advances in CALIPSO (IIR) cirrus cloud properties retrievals — Part 1: Method and testing" by Mitchell et al.

The manuscript presents significant improvements to the retrieval method previously described by Mitchell et al. (2018). These include methodological refinements, incorporation of additional in-situ observations to develop new empirical relationships linking retrieval properties with CALIPSO observations, and benefits from recent improvements in the IIR and CALIOP operational products (version 4). The updated retrieval approach appears not only to show more precise results but also notably enhances sensitivity to tropical tropopause layer cirrus and polar stratospheric clouds.

The method presented, following Mitchell et al. (2018), remains unique and built upon robust theoretical considerations. A major advantage over existing retrieval methods is that it does not rely on assumptions regarding the ice particle size distribution. It provides an extensive set of (interconnected) microphysical properties, such as the IWC, effective size De, extinction, and Ni. The latter is particularly valuable to the community, given the limited availability of such retrievals from satellite remote sensing.

The authors provide thorough uncertainty evaluations, including the propagation of retrieval uncertainties and statistical comparisons with in-situ observations. I have no fundamental criticisms of the methodology itself. The study clearly falls within the scope of ACP, although, in my opinion, it could fit more naturally as a technical note.

The retrieval methodology is robust, and the resulting product is highly valuable. While I have few specific concerns regarding the method itself, I do have several major general comments on clarifying the nature of the retrievals, the capabilities and limitations of the method, and aspects of its validation. I recommend publication after addressing the major revisions detailed below.

**General comments:**

1. The authors have thoroughly discussed uncertainties throughout the manuscript, including uncertainties propagated through operational CALIPSO products (sections 2.2.3 and 2.2.4) and indirect comparisons with in-situ measurements. However, the key conclusions related to these uncertainties are, in my opinion, not sufficiently highlighted in the current conclusions section. I suggest including a more comprehensive summary of this uncertainty analysis and explicitly outlining the method's limitations. Additionally, clearly defining optimal conditions for the retrieval's application would help users better understand how to effectively utilize or compare the dataset (e.g., for model evaluations). Specifically, aspects related to land/sea contrasts and impacts from snow and sea ice discussed in section 2, along with clarifications associated with the other points below, should be better highlighted in the conclusions.

2. Operational product uncertainties are adequately discussed, and identified as significant uncertainty sources in section 3.4. However, uncertainties stemming from in-situ-based parameterizations (e.g., regressions in Table 3 and relationships in Table 5) should also be acknowledged and discussed similarly, as they might be even more significant for this method. Section 4 provides climatological comparisons between retrievals and in-situ measurements, mostly confirming that in-situ measurements fall within the retrieval spread, except notably for Ni in tropical regions (Figures 17 and 18). I think the manuscript would greatly benefit from one or two detailed case studies, involving coincident satellite and in-situ observations (covering both tropical and mid-latitude scenarios). Such case studies should be feasible using the campaigns already discussed or the Krämer et al. (2020) dataset used for validation.

3. It would also be beneficial to clarify what exactly the retrieved quantities represent. The IIR measurements inherently reflect vertically integrated quantities, weighted by emissions from various cloud layers. Section 2.2.5 briefly discusses these aspects in terms of equivalent layer thickness, but the implications for interpreting retrievals remain somewhat unclear. Typically, passive sensors retrieve integrated properties (IWP, optical depth), while this method retrieves parcel-scale properties (IWC, Ni, De, extinction). Clarifying whether the retrievals represent the entire cloud vertical extent or a specific altitude associated with the radiative temperature (Tr) is needed for proper user interpretation. Comparative case studies as mentioned in my comment #2 could involve CALIOP or DARDAR vertical profiles to clarify these points.

4. The retrieval method relies primarily on parameterizations derived from tropical campaigns, with only one mid-latitude campaign (SPARTICUS) providing somewhat less robust data due to reliance on 2D-S measurements and subsequent corrections. The question of the global applicability of the method should be addressed more explicitly, particularly in the conclusions, given the global applications discussed in Part 2 of this series. Clarifying whether significant limitations exist, such as for high-latitude regions, and advising users on potential constraints would be valuable.

5. It would be informative to directly compare the performance of this updated retrieval method with Mitchell et al. (2018), perhaps through global maps or temperature-dependent analyses. Such comparisons would more clearly highlight the specific advancements made, aligning directly with the manuscript's title.

**Specific comment**

1. Given the extensive general comments, I have only one concise specific comment or rather a question: Section 2.2.5 illustrates the utility of the IIR weighting function to reconstruct CALIOP extinction profiles. Could similar reconstructions be feasible for IWC, De, and Ni profiles? While this clearly extends beyond the present study's scope, briefly mentioning this possibility as a potential future improvement in the conclusions would be beneficial.

---

## Author Comment (AC1)

**Author responses to Reviewer 1**

Format: The reviewers' comments are in black font while author responses are in red font. Text in red font italics indicates revised/added text in the revised manuscript.

We understand that reviewing this paper took a lot of time and effort, and we sincerely thank you for your comments that have improved this paper. Below are our responses to the general and specific comments:
* * *
The manuscript by Mitchell et al. improves upon the development of an earlier methodology published in 2018 to retrieve the layer properties of optically thin cirrus clouds using a combination of the IIR and CALIOP instruments on the CALIPSO satellite. The synergistic combination of lidar and infrared measurements is a unique property of the CALIPSO satellite and were initially conceived for the very purpose of retrieving cirrus microphysics. The proposed algorithm developed in this manuscript seeks to use the ratio of the 12 micron to 10 micron radiances to derive a relationships between absorption optical depth and microphysics and the vertically integrated attenuated lidar backscatter that is related to the visible optical depth at 523nm to derive the layer-mean ice water content, the effective particle size, and ice crystal number concentration, Ni. The retrieval is formulated using a closed set of simplified equations for the radiative properties. The closure is achieved using analysis of in situ cirrus particle size distribution data from multiple cirrus measurement campaigns along with assumptions regarding the absorption properties of the PSD. The authors compare statistical results of the retrieval algorithm (i.e. regionally derived cirrus properties) to the statistics of the cirrus properties in the campaigns. Overall, I find the methodology unique and innovative builds on a long history of cirrus data analysis by the authors. However, I have several major concerns that I think the authors must address in a major revision of the present manuscript.

1. The algorithm is dependent upon in situ cirrus data particle size distributions for relationships that allow for closure of the algorithm equations. While the general approach seems sound, there are several critical issues that should be addressed in more detail. The first of these is the fact that the algorithm retrieves the properties of a layer that may be several kilometers deep where vertical variations in cirrus properties are often considerable over the depth of the layer. However, aircraft data are typically collected along level flight lines within cirrus layers and they do not typically sample the vertical structure. Assuming that the relationships developed from the in situ data are relevant in small volumes, why would the relationships apply to entire cirrus layers particularly when the infrared absorption properties are likely weighted to a different region of the layer than the visible optical depth?

   The relationships derived from in situ data relate $\beta_{eff}12/10$ to $N_i/A_{PSD}$, $N_i/IWC$, and $1/Q_{abs,eff}$ (12 µm). $N_i/A_{PSD}$ is the inverse of an area (units are $cm^{-2}$ in Fig. 14), $N_i/IWC$ is the inverse of a mass (units are $g^{-1}$ in Fig. 14). $1/Q_{abs,eff}$ (12 µm) has no unit and $\beta_{eff}12/10$ is a ratio of effective absorption optical depths. There is no notion of volume involved, which justifies applying the relationships to layers.

   We show in Sect. 2.2.5 how we use the CALIOP extinction profile in the visible to estimate the IIR weighting function. In panel a where the cloud emissivity at 12.05 µm is small (0.06), the IIR weighting function and the CALIOP extinction profiles have very similar shape. In panel b, cloud emissivity is larger (0.44) and we see that relative

to the CALIOP extinction profile, there is more weight at 12.05 µm in the upper part of the cloud and less in the lower part. The inferred IIR equivalent layer thickness represents the portion of the cloud contributing to the absorption optical depth. The cloud properties such as extinction ($\alpha_{ext}$), $N_i$ or IWC characterize the layer.

We corrected and clarified the beginning of Section 2.1 (that briefly describes the M2018 retrieval) as follows (changes in italic):

« *Layer* $N_i$ is derived from the $N_i$/IWC ratio after retrieving *layer* IWC from $\alpha_{ext}$ and the empirical $D_e$ - $\beta_{eff}$ relationship. The uncertainty in $N_i$ can be reduced by eliminating its dependence on the empirical $D_e$ - $\beta_{eff}$ relationship and by replacing the $N_i$/IWC ratio with the $N_i$/$A_{PSD}$ ratio, where $A_{PSD}$ is the PSD projected area *per unit volume* directly measured by the 2D-S probe (Lawson et al, 2006; Lawson, 2011). »

2. Continuing with the previous comment, the PSDs measured in cirrus by probes are well known to contain shattering artifacts.While all modern probes now contain tips that reduce shattering and data are analyzed to remove clusters of small particles that likely resulted in shattering, this effect is still present and likely dominates (or at least biases) the number concentration of ice crystals, Ni, in aircraft measurements. I note that Kramer et al. (2020) in their cirrus climatology spend a large section of their analysis on this topic and still feel that it biases their results. The PSDs shown in Figure 8 in this manuscript are representative where there is a main mode of larger particles that is in the several hundred-micron size range and then tails off smoothly toward smaller sizes (like a modified gamma distribution would do) but this mode is interrupted in the 10's of microns range by a smaller mode that occupies the smallest several size bins.  Can the authors explain the microphysical mechanism that would result in this behavior?  In my experience, this mode of small particles is omnipresent in cirrus measurements, and, in my opinion, it likely represents shattering artifacts that have not been mitigated by tips or removed through software.  I'd be happy if the authors could explain to me why I am wrong about this.  Can the authors point to in situ data that does not show this small mode?  However, it remains the case that this residual small mode biases the Ni in their data so the presence of the small mode should be addressed either via explaining the microphysical processes that produce it or by finding some way of removing it that is more sophisticated than simply dividing it by 10.4 as described in the paper that they apply to the first bin of the 2DS data.

While the PSDs shown in Fig. 8 are typical, there are many examples where the PSD small mode (peaking ~ 2 microns) is absent (i.e., the concentrations of small crystals with size exhibit a "flat" behavior).  And there are other examples where the PSD tends to be unimodal, without a distinct small mode.  In these cases, Ni tends to be anomalously high, suggesting homogeneous freezing nucleation (hom) dominates for such PSDs.  An example from the ATTREX campaign at -80 °C is shown below:

[Figure]

GlobalHawk 21:22:45-21:23:25

Automatic Mean PSD
Reff (A) ≈ 9.18 μm
Conc ≈ 1475 #/L
LWC ≈ 0.001 g/m³
IWC (BF95) ≈ 0.001 g/m³
dBz ≈ 0

Hawkeye FCDP R1
Hawkeye 2DS R1
Mean

Here it is seen that ice particle sizes are less than 60 μm, making shattering very unlikely due to the blocky, simple shapes of the crystals and their sizes. When the PSD is unimodal and Ni exceeds 200 L⁻¹ (Ni = 1475 L⁻¹ in this case), this is typical of hom. Hence, hom can be an abundant source of small ice crystals and may contribute to the PSD small mode. That is, a pronounced small mode is not necessarily an indication of ice artifacts from shattering. The details of how turbulence-induced transient hom events affect the PSD are addressed in Kärcher et al. 2025, https://doi.org/10.1038/s41612-025-01024-w.

3. The authors address algorithm uncertainty in an appendix by deriving a number of equations that propagate the error in the measurements through to the retrieved quantities. While this seems thorough enough, they do not present actual results of the uncertainty analysis – just the equations. They do mention that pixel to pixel uncertainty in Ni can be on the order of a factor of 2, they do not, however, present an actual analysis of the error or show how it depends upon the uncertainties in the input variables or assumptions. Such an analysis is a requirement in my opinion.

As noted by the reviewer, the uncertainties are estimated by propagating the error in the IIR measurements to the retrieved quantities. Uncertainties in the X-$\beta_{eff}$ relationships are not included. This was clarified by adding the following sentence in Sect. 3.4 :

« *Note that additional uncertainties in the X- $\beta_{eff}$ relationships are difficult to estimate and are not included in this assessment.* ».

An analysis of errors is provided in Table 6 for the ATTREX campaign and in Table S1 in the Supplement for the POSIDON campaign. These uncertainties are discussed in Sect. 4.1, as shown below :

« Table 6 shows median retrieved properties and relative uncertainty estimates for ATTREX for cirrus having ~ 0.2 - 0.3 < τ < ~ 3 (left) and ~ 0.01 < τ < ~ 3 (right). A similar table for the POSIDON campaign is shown in the Supplement (Table S1). In Table 6, median Δβ$_{eff}$ ranges from 0.03 to 0.44 where median τ = 0.05 at $T_r$ = 193 K, ΔNi/Ni = 1.88 and ΔD$_e$/D$_e$ = 0.98. The smallest median ΔN$_i$/N$_i$ is 0.35 at $T_r$ = 193 K when only the thicker clouds are sampled and median τ is somewhat small (0.23), but this is compensated by the fact that β$_{eff}$ = 1.57 where the sensitivity of the technique is very favorable. In contrast, median β$_{eff}$ is 1.056-1.058 at 233 K, where the sensitivity of the technique is less favorable, which explains the occurrence of relative uncertainties larger than 2.4 despite the small Δβ$_{eff}$ = 0.03.

Again, these uncertainty estimates characterize random uncertainties of individual retrievals and are reduced for statistical analyses involving a large number of samples. »

Uncertainties are shown and discussed in the case study which is now added at the end of Sect. 4.2 (see comment #4).

4. The authors move from algorithm development directly to statistical comparisons with other campaigns. They do not show actual data collected by the sensors and the retrieval results with error bars that would result from application of their algorithm to actual cirrus layers. Furthermore, it seems necessary to demonstrate an actual case or two in circumstances when in situ aircraft data were being collected data underneath the satellites. The SPartICus data that is used by the authors in their analysis has approximately two dozen flights where the Lear Jet flew along the paths of the CloudSat and CALIPSO satellites. These are well documented in Deng et al. (2013; *DOI: 10.1175/JAMC-D-12-054.1*). As a matter of fact, since the authors use SPartICus data, it seems reasonable for them to replicate the results in Deng et al. This would be quite straightforward. The authors also use data collected in TC4. Mace et al. (2010; *doi:10.1029/2009JD012517*) illustrate a case when the NASA DC8 flew along the CloudSat and CALIPSO tracks in tropical cirrus. Replicating these direct comparisons between algorithm results and in situ aircraft data is a necessary step toward establishing confidence in the algorithm results and would go a long way toward addressing many of the earlier concerns I have raised.

We move from algorithm development to statistical comparisons with in situ observations because comparing layer retrievals and in situ observations is not straightforward, but we agree that this should be illustrated.

We added a case study during the SPARTICUS campaign at the end of Sect. 4.2 for 30 March 2010. The case studies shown in Deng et al. (2013) could not be used because the scenes were not adpated for our IIR retrievals for the following reasons. The cirrus is prevailingly opaque to CALIOP for the thick-anvil case on 17 April 2010 and the thick-cirrus case on 12 June 2010. The thin-cirrus case on 22 April 2010 could not be used because low clouds were present below 5-km altitude. For the 1 April 2010 case, a portion of the cloud is a single-layer semi-transparent ice cloud but the radiative temperature is most of the time warmer than 235 K.

Below is the case study and the new text added at the end of Sect. 4.2 :

   *The difficulty to directly compare IIR layer retrievals and aircraft in situ data is illustrated in the SPARTICUS case study shown in Fig. 21 for 30 March 2010. Following the CALIPSO track, the Learjet flew northwards (leg 1, triangles) with measurements at 11 km altitude 7 to 3 minutes before the CALIPSO overpass and then southwards (leg 2, diamonds) with measurements at 11.6 km altitude 6.5 to 8 minutes after. CALIPSO detected a single layer cirrus of top altitude near 12.6 km. The colors in panel a represent the altitude-dependent CALIOP extinction profiles scaled to IIR $\tau$. The colors inside the triangles and diamonds indicate the PSD extinctions larger than 0.01 $km^{-1}$ after averaging over a 30-s period. At the top of panel a is IIR layer $\alpha_{ext}$, which was derived from $\tau$ and $\Delta z_{eq}$ shown in panel b. As discussed in Sect. 2.2.5, $\Delta z_{eq}$ represents the portion of a layer contributing the most to the cloud emissivity. The solid black line in panel a is the radiative altitude corresponding to $T_r$, which to a first approximation corresponds to the mid cloud altitude (see Fig. 7). IIR $D_e$ in red in panel c (with vertical bars indicating $D_e \pm \Delta D_e$) is lower than the in situ values, which is explained by the fact that both legs were below the radiative altitude. That is, in the lower half of an ice cloud, mean ice particle size tends to be larger and $N_i$ lower relative to the*

[Figure]

*Figure 21. Comparison of IIR retrievals and in situ observations on 30 March 2010 during the SPARTICUS field campaign (CALIPSO granule 2010-03-30T19-27-25ZD). (a) extinction profile derived from the CALIOP lidar, IIR layer $\alpha_{ext}$, and PSD extinctions in leg 1 (triangles) and leg 2 (diamonds). The stars denote the boundaries of the CloudSat radar GEOPROF cloud mask; (b) IIR $\tau$ (red) and $\Delta z_{eq}$ (black, right axis); (c) IIR (red) and in situ (triangles and diamonds) $D_e$; (d) same as (c) but for $N_i$. The vertical bars in red in panels b-d represent the IIR estimated uncertainties.*

*upper half due to diffusional growth and aggregation (e.g., Mitchell, 1988; 1994; Field and Heymsfield, 2003). Only a lower portion of the cloud was detected by the CloudSat radar (shown by the stars) between latitudes 36.5° and 36.78°. IIR $D_e$ is the smallest (around 20 µm) south of 36.5° and north of 36.78° where there is no radar detection, indicating crystals smaller than about 40 µm. Moreover, the absence of radar detection outside this CloudSat domain (defined by the stars) indicates ice particles smaller than ~ 40 µm, revealing a vertical gradient in ice particle size. Regarding $N_i$ (panel d showing $N_i \pm \Delta N_i$), the large IIR $N_i$ values in red between 300 and 850 $L^{-1}$ are explained by large $N_i$ near cloud top. Regarding uncertainties, $\Delta N_i$ is overall equal to about 80 $L^{-1}$ and its noticeable increase up to 300 $L^{-1}$ in the northernmost part of the cloud is due to the decrease of $\tau$. The same observation applies to $\Delta D_e$ which is between 3 and 11 µm. To summarize, while the vertically resolved extinction retrievals exhibit reasonable agreement with the in situ extinction measurements, the bulk cloud layer retrievals often do not exhibit similar agreement, and this appears to be due to vertical gradients in $D_e$ and $N_i$ and aircraft sampling location. This case study has been classified as ridge crest cirrus which have higher $N_i$ than the other cirrus cloud classes described in Muhlbauer et al. (2014). In this regard, the retrievals here are consistent with this category of cirrus cloud.*

---

## Author Comment (AC2)

**Author responses to Reviewer 2**

Format: The reviewers' comments are in black font while author responses are in red font. Text in red font italics indicates revised/added text in the revised manuscript.

We understand that reviewing this paper took a lot of time and effort, and we sincerely thank you for your comments that have improved this paper. Below are our responses to the general and specific comments:
* * *
Review of "Advances in CALIPSO (IIR) cirrus cloud properties retrievals — Part 1: Method and testing" by Mitchell et al.

The manuscript presents significant improvements to the retrieval method previously described by Mitchell et al. (2018). These include methodological refinements, incorporation of additional in-situ observations to develop new empirical relationships linking retrieval properties with CALIPSO observations, and benefits from recent improvements in the IIR and CALIOP operational products (version 4). The updated retrieval approach appears not only to show more precise results but also notably enhances sensitivity to tropical tropopause layer cirrus and polar stratospheric clouds. The method presented, following Mitchell et al. (2018), remains unique and built upon robust theoretical considerations. A major advantage over existing retrieval methods is that it does not rely on assumptions regarding the ice particle size distribution. It provides an extensive set of (interconnected) microphysical properties, such as the IWC, effective size De, extinction, and Ni. The latter is particularly valuable to the community, given the limited availability of such retrievals from satellite remote sensing. The authors provide thorough uncertainty evaluations, including the propagation of retrieval uncertainties and statistical comparisons with in-situ observations. I have no fundamental criticisms of the methodology itself. The study clearly falls within the scope of ACP, although, in my opinion, it could fit more naturally as a technical note. The retrieval methodology is robust, and the resulting product is highly valuable. While I have few specific concerns regarding the method itself, I do have several major general comments on clarifying the nature of the retrievals, the capabilities and limitations of the method, and aspects of its validation. I recommend publication after addressing the major revisions detailed below.

**General comments:**

1. The authors have thoroughly discussed uncertainties throughout the manuscript, including uncertainties propagated through operational CALIPSO products (sections 2.2.3 and 2.2.4) and indirect comparisons with in-situ measurements. However, the key conclusions related to these uncertainties are, in my opinion, not sufficiently highlighted in the current conclusions section. I suggest including a more comprehensive summary of this uncertainty analysis and explicitly outlining the method's limitations. Additionally, clearly defining optimal conditions for the retrieval's application would help users better understand how to effectively utilize or compare the dataset (e.g., for model evaluations). Specifically, aspects related to land/sea contrasts and impacts from snow and sea ice discussed in section 2, along with clarifications associated with the other points below, should be better highlighted in the conclusions.

Item (1) in the conclusion was modified and now reads (changes in italic):

"By expanding the sampling range to include optically thinner cirrus clouds (0.01 < τ < 3) *over oceans*, the sampling has become more representative of all cirrus clouds *over oceans*. *The sampling over land, snow,*

*and sea ice remains limited to thicker cirrus clouds having $\tau > 0.3$ because of larger random uncertainties in IIR absorption optical depth retrievals.*"

2. Operational product uncertainties are adequately discussed and identified as significant uncertainty sources in section 3.4. However, uncertainties stemming from in-situ-based parameterizations (e.g., regressions in Table 3 and relationships in Table 5) should also be acknowledged and discussed similarly, as they might be even more significant for this method. Section 4 provides climatological comparisons between retrievals and in-situ measurements, mostly confirming that in-situ measurements fall within the retrieval spread, except notably for Ni in tropical regions (Figures 17 and 18). I think the manuscript would greatly benefit from one or two detailed case studies, involving coincident satellite and in-situ observations (covering both tropical and mid-latitude scenarios). Such case studies should be feasible using the campaigns already discussed or the Krämer et al. (2020) dataset used for validation.

We added the following sentence in Sect. 3.4: "*Note that additional uncertainties in the $X$- $\beta_{eff}$ relationships are difficult to estimate and are not included in this assessment.*"

Following also the recommendation by reviewer #1, a case study during the SPARTICUS campaign is now added at the end of Sect. 4.2, which now reads:

*"The difficulty to directly compare IIR layer retrievals and aircraft in situ data is illustrated in the SPARTICUS case study shown in Fig. 21 for 30 March 2010. Following the CALIPSO track, the Learjet flew northwards (leg 1, triangles) with measurements at 11 km altitude 7 to 3 minutes before the CALIPSO overpass and then southwards (leg 2, diamonds) with measurements at 11.6 km altitude 6.5 to 8 minutes after. CALIPSO detected a single layer cirrus of top altitude near 12.6 km. The colors in panel a represent the altitude-dependent CALIOP extinction profiles scaled to IIR $\tau$. The colors inside the triangles and diamonds indicate the PSD extinctions larger than 0.01 $km^{-1}$ after averaging over a 30-s period. At the top of panel a is IIR layer $\alpha_{ext}$, which was derived from $\tau$ and $\Delta z_{eq}$ shown in panel b. As discussed in Sect. 2.2.5, $\Delta z_{eq}$ represents the portion of a layer contributing the most to the cloud emissivity. The solid black line in panel a is the radiative altitude corresponding to $T_r$, which to a first approximation corresponds to the mid cloud altitude (see Fig. 7). IIR $D_e$ in red in panel c (with vertical bars indicating $D_e \pm \Delta D_e$) is lower than the in situ values, which is explained by the fact that both legs were below the radiative altitude. That is, in the lower half of an ice cloud, mean ice particle size tends to be larger and $N_i$ lower relative to the upper half due to diffusional growth and aggregation (e.g., Mitchell, 1988; 1994; Field and Heymsfield, 2003). Only a lower portion of the cloud was detected by the CloudSat radar (shown by the stars) between latitudes 36.5° and 36.78°. IIR $D_e$ is the smallest (around 20 µm) south of 36.5° and north of 36.78° where there is no radar detection, indicating crystals smaller than about 40 µm. Moreover, the absence of radar detection outside this CloudSat domain (defined by the stars) indicates ice particles smaller than ~ 40 µm, revealing a vertical gradient in ice particle size. Regarding $N_i$ (panel d showing $N_i \pm \Delta N_i$), the large IIR $N_i$ values in red between 300 and 850 $L^{-1}$ are explained by large $N_i$ near cloud top. Regarding uncertainties, $\Delta N_i$ is overall equal to about 80 $L^{-1}$ and its noticeable increase up to 300 $L^{-1}$ in the northernmost part of the cloud is due to the decrease of $\tau$. The same observation applies to $\Delta D_e$ which is between 3 and 11 µm. To summarize, while the vertically resolved extinction retrievals exhibit reasonable agreement with the in situ extinction measurements, the bulk cloud layer retrievals often do not exhibit similar agreement, and this appears to be due to vertical gradients in $D_e$ and $N_i$ and aircraft sampling location. This case study has been classified as ridge crest cirrus which have higher $N_i$ than the other cirrus cloud classes described in Muhlbauer et al. (2014). In this regard, the retrievals here are consistent with this category of cirrus cloud.*"

[Figure]

*Figure 21. Comparison of IIR retrievals and in situ observations on 30 March 2010 during the SPARTICUS field campaign (CALIPSO granule 2010-03-30T19-27-25ZD). (a) extinction profile derived from the CALIOP lidar, IIR layer $\alpha_{ext}$, and PSD extinctions in leg 1 (triangles) and leg 2 (diamonds). The stars denote the boundaries of the CloudSat radar GEOPROF cloud mask; (b) IIR $\tau$ (red) and $\Delta z_{eq}$ (black, right axis); (c) IIR (red) and in situ (triangles and diamonds) $D_e$; (d) same as (c) but for $N_i$. The vertical bars in red in panels b-d represent the IIR estimated uncertainties.*

3. It would also be beneficial to clarify what exactly the retrieved quantities represent. The IIR measurements inherently reflect vertically integrated quantities, weighted by emissions from various cloud layers. Section 2.2.5 briefly discusses these aspects in terms of equivalent layer thickness, but the implications for interpreting retrievals remain somewhat unclear. Typically, passive sensors retrieve integrated properties (IWP, optical depth), while this method retrieves parcel-scale properties (IWC, Ni, De, extinction). Clarifying whether the retrievals represent the entire cloud vertical extent or a specific altitude associated with the radiative temperature (Tr) is needed for proper user interpretation. Comparative case studies as mentioned in my comment #2 could involve CALIOP or DARDAR vertical profiles to clarify these points.

We clarified as suggested and modified Section 2.2.5 as follows:

1)Title was changed to "*IIR equivalent layer thickness, $\Delta z_{eq}$ and radiative temperature*".

2)The first paragraph now reads: "*Even though the IIR is a passive instrument that retrieves layer integrated quantities such as cloud optical depth, the cloud boundaries information provided by CALIOP allows one to retrieve vertically resolved layer properties such as the layer extinction coefficient. However, the high sensitivity of CALIOP to cloud detection and the expected variability of extinction*

*within the layer are such that only a portion of the cloud layer detected by CALIOP is "seen" by IIR. Thus, relevant for our retrievals is the IIR equivalent layer thickness, $\Delta z_{eq}$, which is estimated using the IIR in-cloud weighting function derived from the in-cloud 532 nm CALIOP extinction profile of vertical resolution, $\delta z$ (Garnier et al., 2021a).*

3)Later in the section, we modified the text to clarify why the red and black curves have similar shape in panel a of Fig. 5 (new text in italic): "The first example in panel a is a TTL cirrus between 15.13 and 16.5 km observed in June 2010. Retrieved $\varepsilon_{eff}$ is equal to 0.06 and the black and red curves have an almost identical shape *because the attenuation term in Eq. (12) is close to 1*".

4)The beginning of the last paragraph was re-written to note that $T_r$ can be seen as a first approximation as the temperature corresponding to mid-cloud optical depth. It now reads: "*The examples shown in Fig. 5 illustrate that the IIR weighting function is in first approximation the CALIOP extinction profile normalized to the optical depth if the attenuation term in Eq, (12) is supposed to be close to 1 and $\varepsilon(i)$ is approximated to the corresponding $\tau_{abs}$ in Eq. (13). This IIR weighting function is also used to determine the cloud centroid radiance and the corresponding radiative temperature, $T_r$, (Garnier et al., 2021a), which is given in red in each panel. The temperatures in black are $T_{top}$ and $T_{base}$ at the layer top and base altitudes, respectively. Because computing a centroid temperature would yield a temperature differing by less than a few tenths of a degree Kelvin (M2018), $T_r$ can be seen as the temperature dividing the cloud optical depth into equal parts.*"

4. The retrieval method relies primarily on parameterizations derived from tropical campaigns, with only one mid-latitude campaign (SPARTICUS) providing somewhat less robust data due to reliance on 2D-S measurements and subsequent corrections. The question of the global applicability of the method should be addressed more explicitly, particularly in the conclusions, given the global applications discussed in Part 2 of this series. Clarifying whether significant limitations exist, such as for high-latitude regions, and advising users on potential constraints would be valuable.

The following sentences were added to the conclusion:

1)We added a new item (3) which reads: "*The computation of in situ $\beta_{eff}$ used in the $X$ - $\beta_{eff}$ relationships was improved using mass-dimension relationships that appear more realistic.*"

2) For item (6), we added "*The $X$ - $\beta_{eff}$ relationships for the SPARTICUS synoptic and the TC4 anvil cirrus yield similar $N_i$ retrievals (see Fig. 17).*"

*3)* The second paragraph under Conclusions now begins with:"*This study should be extended to more field campaigns, in particular at high latitude, to further investigate the variability in the $X$ - $\beta_{eff}$ relationships, which seems more important for $D_e$ than for $N_i$.*"

5. It would be informative to directly compare the performance of this updated retrieval method with Mitchell et al. (2018), perhaps through global maps or temperature-dependent analyses. Such comparisons would more clearly highlight the specific advancements made, aligning directly with the manuscript's title.

We agree that this discussion was missing. The manuscript was modified as follows:

1)Sect. 3.1 now also discusses the impact of using the EM2016 mass-dimension relationship.

- The title of this section is now: "Correction of the smallest bin of the 2D-S probes *and mass-dimension relationship*".

- Figure 13 now includes the relationship for assumption (1), i.e. $N(D)_1$ unmodified, as derived in M2018 to illustrate that using the EM2016 relationship reduces $N_i$.

[Figure]

- The first paragraph now ends with: "*In addition, based on the findings presented in Sect. 2.4, we now use the EM2016 mass-dimension relationships*".

- The beginning of the second paragraph reads (changes in italic): "It is instructive to examine the impact of *these changes* in terms of the $N_i$ retrieval as described in Eq. (2). The field campaign dependence (and thus the 2D-S probe dependence) of $N_i$ enters through the $\beta_{eff}$ dependent terms in Eq. (2), namely through the $N_i/A_{PSD}$ - $\beta_{eff}$ and the $[1/Q_{abs,eff} (12 \mu m)]$ - $\beta_{eff}$ relationships. From Eq. (2), the product of these two ratios is $N_i/\alpha_{abs}$ which is plotted in Fig. 13, showing the impact *of the mass-dimension relationships and of the* $N(D)_1$ assumption on the $N_i$ retrieval. There are three assumptions: (1) $N(D)_1$ is unmodified, meaning the $N(D)_1$ measurement is correct, (2) $N(D)_1$ is modified, divided by 10.4 as discussed in Sect. 2.3, and (3) $N(D)_1 = 0$. These three assumptions were evaluated in Fig. 13 using 2D-S PSD data from the SPARTICUS field campaign measured at temperatures less than -38°C *using the EM2016 relationships. Assumption (1) as derived in M2018 is also shown in black, showing that using the EM2016 relationships (in purple) reduces retrieved $N_i$.*"

2) A new Section 3.5 entitled "Comparison with previous work" was added, which reads:

*For comparison with the previous work (M2018), Figure 17a shows the $N_i /\alpha_{abs}$ ratio from the ATTREX-POSIDON (black), SPARTICUS (navy blue), and TC4 (red) relationships developed in this study vs. the $N_i /\alpha_{abs}$ ratio from SPARTICUS $N(D)_1$ unmodified established in M2018, which, out of the 4 formulations examined in M2018, yielded the largest $N_i$ values (Fig. 5 in M2018). Also shown in Fig. 17a is TC4 $N(D)_1 = 0$ from M2018 (dashed orange) which yielded the lowest $N_i$ values. We see that $N_i$ from this study is about half $N_i$ from M2018 SPARTICUS unmodified for both SPARTICUS and TC4 which are similar to M2018 $N(D)_1 = 0$, while ATTREX-POSIDON is half to two third. Panel b in Fig. 17 compares the $D_e$ retrievals.*

[Figure]

*Figure 17. Comparison of (a) $N_i$ and (b) $D_e$ retrievals in this study (black: ATTREX-POSIDON, navy blue: SPARTICUS, red: TC4) and from TC4 $N(D)_1 = 0$ in M2018 (dashed orange) with retrievals from SPARTCUS $N(D)_1$ unmodified in M2018.*

**Specific comment** 1. Given the extensive general comments, I have only one concise specific comment or rather a question: Section 2.2.5 illustrates the utility of the IIR weighting function to reconstruct CALIOP extinction profiles. Could similar reconstructions be feasible for IWC, De, and Ni profiles? While this clearly extends beyond the present study's scope, briefly mentioning this possibility as a potential future improvement in the conclusions would be beneficial.

The IIR weighting function is derived from the CALIOP extinction profiles at 532 nm which does not contain explicit microphysical information. Its application for estimating profiles of IWC, $D_e$, and $N_i$ would require to also know the in-cloud variation of $\beta_{eff}$, which could be inferred from *a priori* assumptions regarding variations of $D_e$ and further constrained by co-located radar observations when available. New text has been added near the end of Conclusions as follows:

*This CALIPSO retrieval provides layer properties based on layer $\beta_{eff}$ and the IIR weighting function derived from the CALIOP extinction profiles at 532 nm. Future work could aim at estimating in-cloud vertical profiles of IWC, $D_e$, and $N_i$. This would require knowledge of the in-cloud variation of $\beta_{eff}$, which could be inferred from a priori assumptions regarding variations of $D_e$ further constrained by co-located CloudSat radar observations when available.*

---

## Author Response (AR2)

**Author responses regarding the second review from Reviewer 1**

Format: The reviewers' comments are in black font while author responses are in red font. Text in red font italics indicates revised/added text in the revised manuscript.

We understand that reviewing this paper took a lot of time and effort, and we sincerely thank you for your comments that have improved this paper. Below are our responses to the general and specific comments:

**Report by Referee #1:**

While I find the paper to be improved, the authors have only marginally addressed most of my critical comments. However, since they did include a case study that is illustrative of the challenges they face with this algorithm, I recommend that the paper could be published with minor revisions if the editor deems appropriate. My fundamental criticism of the work can be summarized in my response to the authors noting how difficult it is to compare in situ data to a "layer" retrieval. I replied: That comparing "layer" retrievals with in situ data is difficult is exactly the point. If it is difficult to show validation, how can it be that the authors can develop a "layer" retrieval of Ni and IWC based on such in situ data? In the following are my responses to the authors' replies to my original comments:

Major Comment 1: My reply to theirs is as follows: In situ data collected over many campaigns show that middle latitude cirrus have a typical structure with high concentrations of small ice near the top and then depositional growth and aggregation towards the middle of the layer followed by a region of sublimation. Anvils have a slightly different vertical structure with size sorting and aggregation resulting in larger particles near the bases. Regardless, Ni and IWC vary strongly in the vertical implying that a "layer Ni" and "layer iwc" is not defined unless a region of the cloud where the observations are most heavily weighted is carefully defined.

After reading this second review from Reviewer 1, we now think that we have a deeper appreciation regarding this fundamental concern from this reviewer. This concern appears to pertain to the  $X - \beta_{eff}$  relationships in Fig. 14, where X is  $N_i/A_{PSD}$ ,  $N_i/IWC$ , or  $1/Q_{abs}(12 \, \mu m)$ . The reviewer correctly describes the typical vertical structure of Ni and IWC in cirrus clouds, especially geometrically thick cirrus. But there appears to be an implicit assumption that these  $X - \beta_{eff}$  relationships depend on this vertical structure (i.e., the variation of X with height), but this is not the case. X, which is sampled from aircraft (i.e.,

calculated from the sampled PSD), can be sampled at any level in the cloud, and from this sampled PSD,  $\beta_{eff}$  is also calculated using the Modified Anomalous Diffraction Approximation (MADA) as described in Sect. 2.3 of M2018, and Eqns. (4) and (5) from M2018. When X =  $1/Q_{abs}(12 \, \mu m)$ , the same is true but now X is calculated more like  $\beta_{eff}$  is calculated.  $\beta_{eff}$  can be viewed as a radiative characterization or microphysical index of the PSD. Despite large environmental differences among samples, the X- $\beta_{eff}$  relationships obtained are relatively tight (i.e., dispersion is not large). This enables them to be used whereby a given point on these X –  $\beta_{eff}$  relationships represents a cloud layer of arbitrary thickness where  $\beta_{eff}$  is related to the PSD. The retrieval then matches the  $\beta_{eff}$  from these in situ X –  $\beta_{eff}$  relationships with the IIR retrieved  $\beta_{eff}$  to obtain retrieved X. Since the IIR retrieved  $\beta_{eff}$  corresponds to the extinction-weighted PSD for the cloud layer, retrieved X corresponds to this extinction-weighted PSD.

**A new second paragraph has been added to Sect. 3.2:**

"Note that X, which is sampled from aircraft (i.e., calculated from the sampled PSD), can be sampled at any level in the cloud, and from this sampled PSD,  $\beta_{\rm eff}$  is also calculated using the Modified Anomalous Diffraction Approximation (MADA) as described in Sect. 2.3 of M2018, and Eqns. (4) and (5) from M2018. When  $X = 1/Q_{\rm abs}(12~\mu{\rm m})$ , the same is true but now X is calculated more like  $\beta_{\rm eff}$  is calculated.  $\beta_{\rm eff}$  can be viewed as a radiative characterization or microphysical index of the PSD. Despite large environmental differences among samples, the X- $\beta_{\rm eff}$  relationships obtained are relatively tight (i.e., dispersion is not large). This enables them to be used whereby a given point on these X- $\beta_{\rm eff}$  relationships represents a cloud layer of arbitrary thickness where  $\beta_{\rm eff}$  is related to the PSD. The retrieval then matches the  $\beta_{\rm eff}$  from these in situ X- $\beta_{\rm eff}$  relationships with the IIR retrieved  $\beta_{\rm eff}$  to obtain retrieved X. Since the IIR retrieved  $\beta_{\rm eff}$  corresponds to the extinctionweighted PSD for the cloud layer, retrieved X corresponds to this extinction-weighted PSD."

Major Comment 2: My reply to theirs is a follows: I agree that the example at -80 in the tropical tropopause cirrus show no evidence of shattering because, I think, there are too few large particles to shatter. A single case study does not address my criticism however. What fraction of insitu data show an insignificant mode of likely shattered particles? For instance, I contend that even in fall streaks of mid-latitude cirrus where no small particles should be physically present still show a bimodal distribution with the small mode contributing significantly to Ni. The SPARTICUS 2DS data frequently show this artifact, albeit of much less amplitude than earlier campaigns, even though anti-shattering tips were used.

The first version of "Major Comment 2" asked "Can the authors explain the microphysical mechanism that would result in this (bimodal) behavior?" and "Can the authors point to in situ data that does not show this small mode?" We have responded to this question by providing an example PSD that does not show this small mode and that also shows strong evidence of homogeneous ice nucleation (henceforth hom), arguing that bimodal PSD may result when both heterogeneous ice nucleation (i.e., het) and hom are active. We cited Kärcher et al. (2025, npj Climate & Atmos. Sci., titled "Dissecting cirrus clouds: navigating effects of turbulence on homogeneous ice formation") for evidence that hom often occurs simultaneously with het, acting to broaden the PSD. Due to limitations in their modeling system, bimodal PSDs were not predicted in Kärcher et al., but the predicted PSD broadening by hom appears consistent with bimodal PSD behavior.

During ATTREX and POSIDON, both the 2DS and the FCDP were used, and they overlap between 5 and 45 microns. With the exception of the 1st size-bin (5 - 15 um) of the 2DS, agreement was generally very good in this overlap region (that should be affected by shattering if it was an issue). If random shattering was strongly biasing N(D) at these sizes, such agreement would be unlikely it seems. Note that the anomalous behavior of the 1st size bin may be unrelated to ice particle shattering based on conversations with Dr. Paul Lawson (who developed the 2DS probe along with others at SPEC, Inc.) and Gurganus and Lawson (2018, J. Atmos. & Oceanic Tech., titled "Laboratory and Flight Tests of 2D Imaging Probes: Toward a Better Understanding of Instrument Performance and the Impact on Archived Data").

Research on ice particle shattering might continue for years to come. However, it is worth pointing out that the ice PSD is a function of multiple physical processes including deposition growth, aggregation, sublimation, the Kelvin effect, and sedimentation. In Jensen et al. (2024, JGR, titled "The Impact of Gravity Waves on the Evolution of Tropical Anvil Cirrus Microphysical Properties"), it was found that "the combination of waves and the Kelvin effect drives growth of crystals with initial diameters of .3–10 µm to sizes of 20–30 µm." This may be one process contributing to bimodal PSDs.

The bimodality seen in fall streaks may be the result of two separate populations of particles produced through hom and het, respectively, falling at different mean velocities of course. The falling ice humidifies the adjacent atmosphere, allowing the smaller ice crystals to survive.

Major Comment 3: My reply to theirs is as follows: That comparing "layer" retrievals with in situ data is difficult is exactly the point. If it is difficult to show validation, how can it be that

the authors can develop a "layer" retrieval of Ni and IWC from such in situ data?

Also, there were several flight lines during SPARTICUS where the Lear Jet flew ramps profiling the cirrus layer. These ramps should reasonably allow the authors to quantify the vertical structure and compare "layer" quantities.

The rationale for developing a layer retrieval for Ni and IWC has been explained in our response to Major Comment 1 above.